# Pan-cancer analysis of mRNA stability for decoding tumour post-transcriptional programs

Gabrielle Perron [1,2], Pouria Jandaghi[2], Elham Moslemi[2], Tamiko Nishimura[2], Maryam Rajaee[2], Rached Alkallas [1,2,3], Tianyuan Lu [1,2,4], Yasser Riazalhosseini [1,2] & Hamed S. Najafabadi [1,2✉]

Measuring mRNA decay in tumours is a prohibitive challenge, limiting our ability to map the post-transcriptional programs of cancer. Here, using a statistical framework to decouple transcriptional and post-transcriptional effects in RNA-seq data, we uncover the mRNA stability changes that accompany tumour development and progression. Analysis of 7760 samples across 18 cancer types suggests that mRNA stability changes are ~30% as frequent as transcriptional events, highlighting their widespread role in shaping the tumour transcriptome. Dysregulation of programs associated with >80 RNA-binding proteins (RBPs) and microRNAs (miRNAs) drive these changes, including multi-cancer inactivation of RBFOX and miR-29 families. Phenotypic activation or inhibition of RBFOX1 highlights its role in calcium signaling dysregulation, while modulation of miR-29 shows its impact on extracellular matrix organization and stemness genes. Overall, our study underlines the integral role of mRNA stability in shaping the cancer transcriptome, and provides a resource for systematic interrogation of cancer-associated stability pathways.

[1] Department of Human Genetics, McGill University, Montreal, QC H3A 1B1, Canada. [2] McGill Genome Centre, Montreal, QC H3A 0G1, Canada. [3] Rosalind and Morris Goodman Cancer Institute, Montreal, QC H3A 1A3, Canada. [4] Quantitative Life Sciences Program, McGill University, Montreal, QC H3A 1E3, Canada. ✉email: hamed.najafabadi@mcgill.ca

Widespread disruption of gene expression programs is a hallmark of cancer and underlies the extensive transformation of tumour cell identity and behavior. Among the least understood aspects of this gene expression remodeling is the regulation of mRNA stability and decay. Previous studies have found specific programs that are involved in tumourigenesis or metastasis through modulation of mRNA stability[1–8]; however, the extent to which mRNA stability contributes to cancer cell transcriptome has not been systematically studied, and the associated regulatory networks are mostly unknown. A key limitation in studying these post-transcriptional programs stems simply from our lack of ability to measure mRNA decay rate in vivo: traditional methods that measure mRNA decay rely on in vitro manipulations such as transcriptional inhibition with chemical inhibitors (e.g. actinomycin D) or metabolic labeling with nucleoside analogues (e.g. 4-thiouridine), combined with time series measurements of transcripts[9–11]. Despite recent improvements[12,13], these methods are resource-intensive, have inherent limitations and biases such as triggering cellular stress and pleiotropic effects[14], and, most importantly, are only applicable to in vitro models. As a result, the mRNA stability landscape of tumour remains almost completely uncharted across different cancer types.

A potential solution comes from recent studies showing that tissue RNA-seq data contain enough information to disentangle transcription rate from mRNA decay rate. Briefly, under the assumption that RNA processing rate is constant[15,16], any change in unspliced (pre-mature) mRNA abundance (estimated from intronic reads) must reflect a proportional change in transcription rate, while any change in spliced (mature) mRNA abundance (estimated from exonic reads) reflects the combined effect of transcription rate and mRNA decay (Fig. 1a). This model enables the estimation of differential mRNA stability based on how the ratio of exonic and intronic reads changes across conditions[15]. A recent improvement on this model generalizes the unspliced-spliced relationship as a power-law function, with the power-law exponent reflecting the coupling between transcription rate and splicing rate[17] (Supplementary Fig. 1a, b).

Here, we build on these methods to obtain a pan-cancer map of mRNA stability changes between tumour and normal tissues, as well as the mRNA stability changes that accompany tumour progression. To do so, we first introduce a general framework for statistical analysis of differential mRNA stability that takes into account the distributional properties of count data. We benchmark this method using experimental measurements of mRNA decay rate, and then apply it to the RNA-seq data from The Cancer Genome Atlas (TCGA) to map the mRNA stability landscapes of 18 cancer types. We identify thousands of transcripts whose stability is altered during tumour formation and/or progression––experimental measurements in cancer cell line models support these findings and suggest a role for mRNA stability alterations in tumour progression and invasiveness. Finally, using network modeling and functional experiments, we identify key microRNAs (miRNAs) and RNA-binding proteins (RBPs) that mediate these changes, providing new insights into the post-transcriptional mechanisms of transcriptome remodelling in cancer.

## Results

### A generalized linear model for statistical testing of mRNA stability
The spliced and unspliced transcripts of each gene follow a power-law relationship, with deviations from this power-law trend reflecting changes in the degradation rate of the mature mRNA[17] (Supplementary Fig. 1a, b). The power-law exponent reflects the coupling between transcription rate and RNA processing rate–an exponent of 1 indicates no coupling between

transcription and processing rate constants, whereas values smaller than 1 indicate that as transcription increases, processing rate constant decreases, potentially due to saturation of the RNA processing machinery (Supplementary Fig. 1a). To use this power-law relationship for the inference of differential stability, it is essential to correctly model the variability in RNA-seq counts. For this purpose, we developed DiffRAC (https://github.com/csglab/DiffRAC), a framework that converts the unspliced-spliced relationship to a generalized linear model whose parameters can then be inferred from sequencing count data using an appropriate error model of choice (Fig. 1b, c and Supplementary Fig. 1c, d).

We evaluated the performance of DiffRAC for estimating differential mRNA stability using a previously published dataset[18,19], consisting of RNA-seq data from mouse embryonic stem cells and terminal neurons, along with experimentally measured transcript half-life measurements after transcriptional blockage with actinomycin D, which here we consider as "ground-truth" measurements for benchmarking purposes. We observed an overall Pearson correlation of 0.22 between RNA-seq-based stability estimates from DiffRAC and ground-truth stability measurements (Fig. 1d and Supplementary Data 1a), in line with previous reports on RNA stability estimation using this specific benchmarking dataset[15,17]. However, for transcripts that had narrow confidence intervals as estimated by DiffRAC, the Pearson correlation between RNA-seq-based estimates and ground truth exceeded 0.5 (Fig. 1d–f), indicating that the confidence intervals estimated by DiffRAC indeed reflect the true uncertainty in estimating differential mRNA stability. Based on (adjusted) $P$ values associated with DiffRAC differential stability estimates, we identified 79 transcripts with higher stability in embryonic stem cells and 37 transcripts with higher stability in terminally differentiated neurons (FDR < 0.05), which closely correspond to differentially stable transcripts based on the ground-truth (Fig. 1g). We performed additional benchmarking using RNA-seq data from NAT10-deficient HeLa cells with matched stability data from metabolic labeling-based BRIC-seq measurements[20]. Using similar analysis methods as those described above, we observed that RNA-seq-based DiffRAC estimates for transcripts with narrow confidence intervals correlate with BRIC-seq stability measurements (Supplementary Fig. 2 and Supplementary Data 1b). Overall, these results suggest that DiffRAC can properly estimate not just the mean differential mRNA stability, but also its uncertainty and statistical significance.

One limitation of the model described above is that, with increasing sample sizes, the number of latent variables that need to be estimated by regression also increases, which can become prohibitively expensive in terms of computational times. To overcome the challenges associated with fitting the model in large sample cohorts, we developed a simplified DiffRAC model that assumes most of the variance in transcription can be explained by the experimental variables (see Methods and Supplementary Fig. 3a–c). This assumption greatly reduces the number of parameters; however, we observed that it does not considerably alter the differential stability estimates in the benchmarking dataset (Supplementary Fig. 3d).

### DiffRAC identifies cancer-associated changes in mRNA stability
To investigate the post-transcriptional changes responsible for transcriptome remodeling in cancer, we performed a pan-cancer analysis of differential mRNA stability across TCGA (The Cancer Genome Atlas, available at https://www.cancer.gov/tcga.), encompassing 7760 samples from 18 cancer types. We used DiffRAC to identify transcripts that were differentially stabilized or destabilized in tumour compared to normal tissues in each cancer type. This analysis revealed an average of 3954 mRNAs

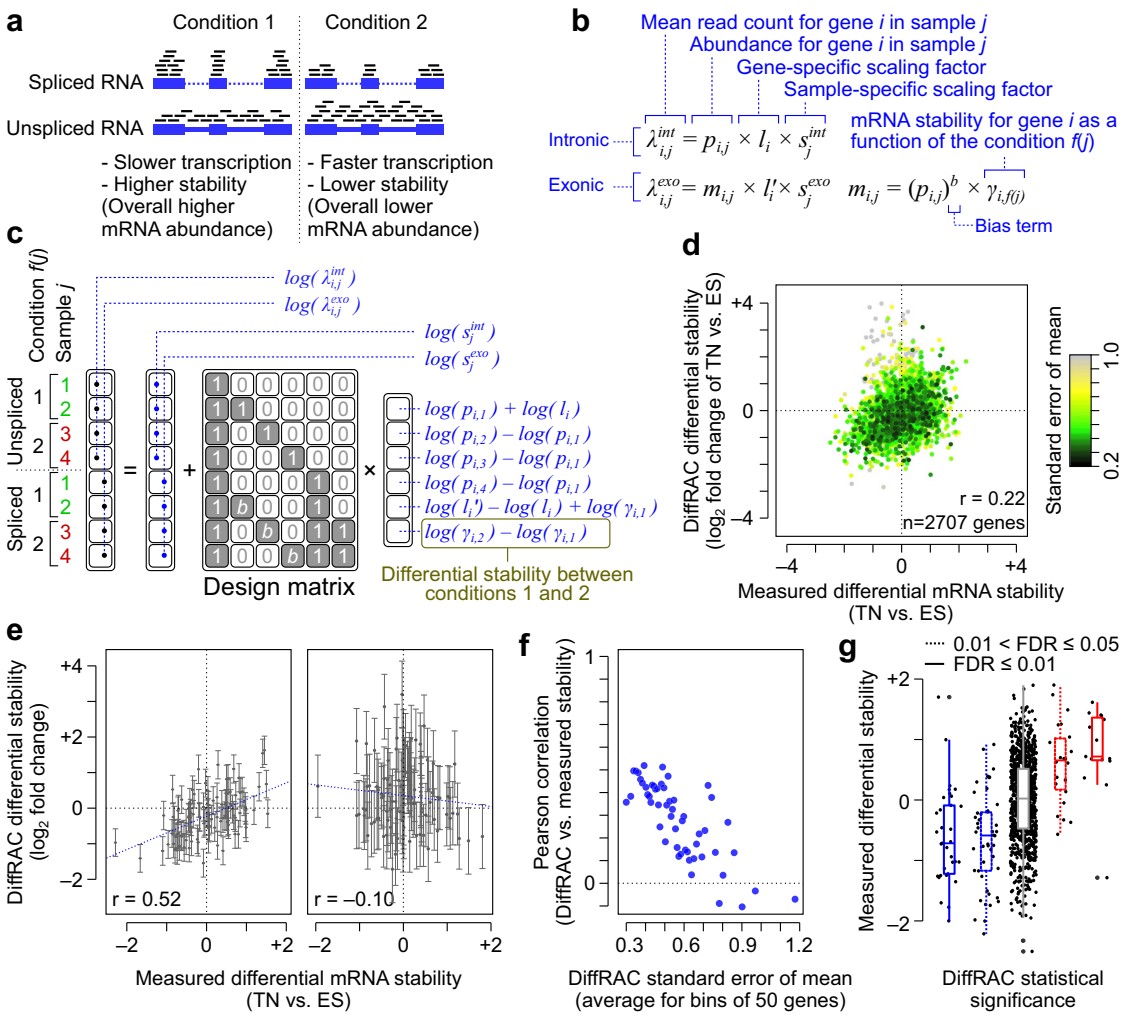

**Fig. 1 Inference of differential mRNA stability using DiffRAC. a** Schematic representation of the effect of transcription and stability on the abundances of unspliced and spliced RNA. **b** DiffRAC models the mean ($\lambda$) of intronic (*int*) and exonic (*exo*) read distribution as a function of pre-mature (*p*) and mature (*m*) transcript abundances, in addition to gene-specific (*l*) and library-specific (*s*) scaling factors. Mature mRNA abundance is modeled as a function of the pre-mature RNA abundance and mRNA stability ($\gamma$), which is in turn a function (*f*) of the experimental variables. Also see Supplementary Fig. 1. **c** An example case with four samples and two experimental conditions, showing how DiffRAC's model can be implemented in a regression with a log-link function, along with the interpretation of regression coefficients (also see Methods). **d** Comparison of DiffRAC stability estimates against experimental mRNA half-life (stability) measurements in mouse ES cells differentiated to terminal neurons (TN)[15,18,19]. Each data point stands for one gene, with the points coloured according the standard error of the mean (SEM) for DiffRAC estimates. **e** Comparison of DiffRAC estimates vs. measured mRNA stability for the 100 genes with the smallest (left) and largest (right) DiffRAC SEMs. Error bars represent the standard error of the mean (SEM). **f** The Pearson correlation between DiffRAC estimates and measured mRNA stability for bins of 50 genes sorted by their SEM. **g** Distribution of experimental mRNA half-life measurements for genes that DiffRAC has identified as significantly destabilized (blue boxplots) or stabilized (red boxplots) in TN vs. ES cells, at FDR cutoffs of 0.05 (dashed line) or 0.01 (solid line). Genes that are not called as significant by DiffRAC are represented with the grey boxplot.

that were differentially stabilized/destabilized per cancer type (FDR-adjusted $p < 0.05$) (Fig. 2a, b, Supplementary Figs. 4 and 5, and Supplementary Data 2), suggesting widespread post-transcriptional remodeling in cancer, with the majority of transcripts showing highly cancer-specific stability profiles (Fig. 2b). Interestingly, across TCGA samples, the degree of stability dysregulation, calculated as the number of differentially stabilized mRNAs per patient, was associated with reduced disease-free survival (log hazard ratio of 0.36, $P < 0.005$, using Cox proportional-hazards model correcting for the confounding effect of patient age, sex, tumour purity and cancer type). Per-cancer-type associations were also mostly positive (Fig. 2c), indicating that a greater disruption of mRNA stability is overall associated with worse patient outcomes.

Several lines of evidence support the reliability of the stability profiles we have inferred. First, we observed that tumour mRNA stability profiles clustered by organ of origin (Fig. 2b), providing an internal validation for the robustness of stability inferences. Secondly, we observed that post-transcriptionally deregulated genes in each cancer type are functionally related (Fig. 2d), consistent with previously reported relationship between post-transcriptional regulons and functional gene modules[21,22]. This analysis also highlights the role of mRNA stability in shaping the functional landscape of the cancer cell. For example, epithelial-mesenchymal transition genes and MYC targets are enriched among stabilized mRNAs across several cancer types, while metabolic pathways such as oxidative phosphorylation and lipid metabolism are highly enriched among destabilized mRNAs, most noticeably in cholangiocarcinoma (CHOL), liver hepatocellular carcinoma (LIHC) and head-neck squamous cell carcinoma (HNSC).

Thirdly, we found that cancer-associated stability changes inferred from tissue RNA-seq data are highly consistent with

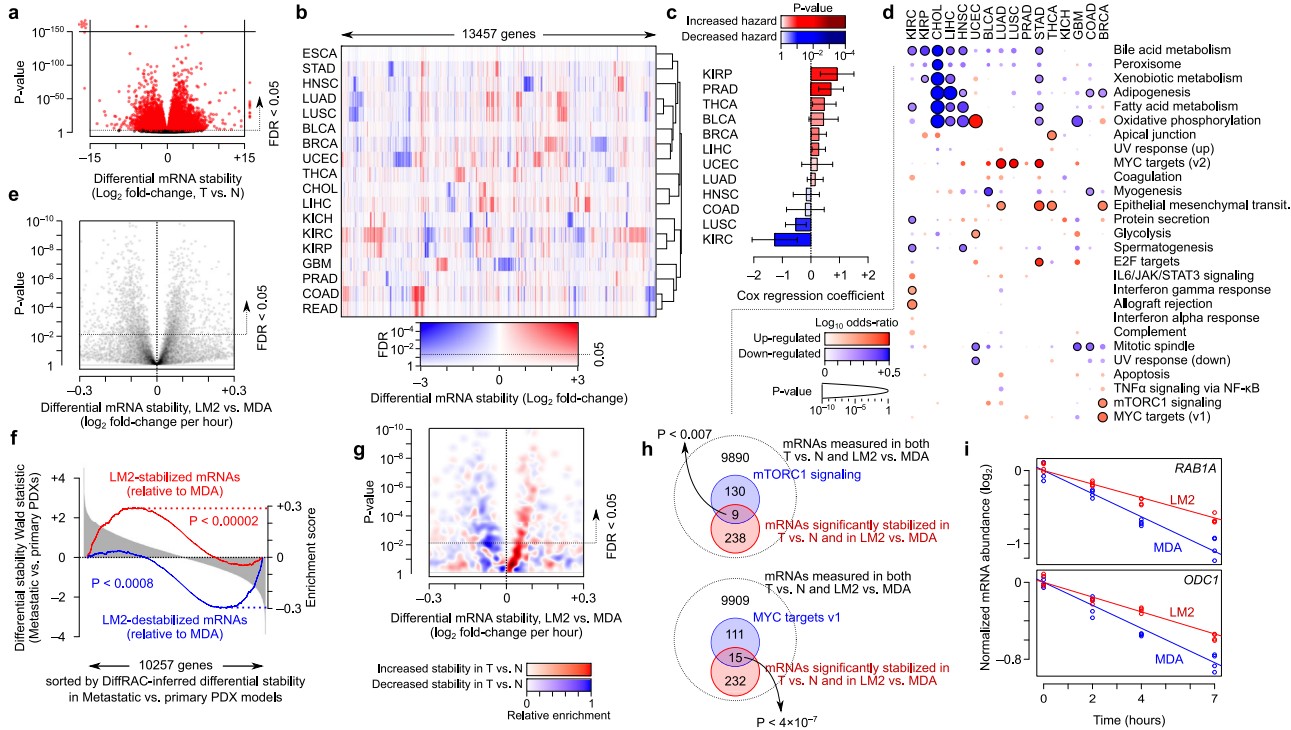

**Fig. 2 Pan-cancer analysis of differential mRNA stability. a** Volcano plot of differential RNA stability between tumour and normal tissues (T vs. N) for 18 TCGA cancer types. See Supplementary Fig. 4 for volcano plots of individual cancer types. **b** Heatmap of differential mRNA stability profiles across TCGA cancers. Genes with significant DiffRAC results in at least one cancer (FDR < 0.05) are included. The colour gradient represents a combination of the $\log_2$ fold-change of mRNA stability and the FDR. BLCA bladder urothelial carcinoma, BRCA breast invasive carcinoma, CHOL cholangiocarcinoma, COAD colon adenocarcinoma, ESCA esophageal carcinoma, GBM glioblastoma multiforme, HNSC head and neck squamous cell carcinoma, KICH kidney chromophobe, KIRC kidney renal clear cell carcinoma, KIRP kidney renal papillary cell carcinoma, LIHC liver hepatocellular carcinoma, LUAD lung adenocarcinoma, LUSC lung squamous cell carcinoma, PRAD prostate adenocarcinoma, READ rectum adenocarcinoma, STAD stomach adenocarcinoma, THCA thyroid carcinoma, UCEC uterine corpus endometrial carcinoma. **c** Associations between the degree of disruption of mRNA stability, defined as a high or low number of differentially stabilized transcripts (relative to the median), and disease-free survival, tested using a Cox proportional-hazards model and correcting for patient age, sex and tumour purity. The bar height represents the Cox regression coefficient, while the colour gradient represents the *p* values, with red representing a worse prognosis, and blue representing a protective effect. The error bars represent the standard error of the mean (SEM). **d** Pathway enrichment analysis of genes with significant differential mRNA stability in each cancer type. Circles with black outline correspond to MSigDB hallmark gene sets that are significantly enriched among cancer-stabilized (red) and cancer-destabilized (blue) mRNAs (FDR < 0.05, Fisher's exact test). Log-odds and *P* values are represented using the colour gradient and circle sizes, respectively. **e** Volcano plot showing the experimentally measured differential stability between highly metastatic MDA-LM2 cell line relative to its parental MDA-MB-231 line (see Methods). **f** Gene set enrichment analysis (GSEA) (Subramanian et al., 2005) for highly metastatic relative to poorly metastatic PDX models of breast cancer[24–26]. Genes (x-axis) are sorted by Wald test statistic of differential stability between metastatic and primary PDXs. The red line represents the enrichment curve for the transcripts that were stabilized in MDA-LM2 relative to MDA-MB-231, while the blue line represents the enrichment curve for the destabilized transcripts. **g** Relative enrichment of transcripts that were stabilized (red) or destabilized (blue) in TCGA-BRCA tumours compared to normal samples, overlaid on the volcano plot from (**e**). Kernel density estimation was used to calculate the density of BRCA-stabilized and destabilized mRNAs across the plot, with the difference between the estimated densities of the two groups shown using the colour gradient. **h** Venn diagrams illustrating the overlap between transcripts that are significantly stabilized in BRCA tumours (relative to normal) and MDA-LM2 (relative to MDA-MB-231), and genes that are part of the mTORC1 signalling (top) or MYC targets (bottom). *P* values are based on Fisher's exact test. **i** Transcription inhibition time-course graphs for two example genes, one involved in mTORC1 signaling (*RAB1A*) and one among MYC targets (*ODC1*). The y-axis shows mRNA abundance after applying variance-stabilized transformation and correcting for mRNA abundance differences between the two cell lines at time zero. Time-course measurements in MDA-LM2 and parental MDA-MB-231 cells are shown in red and blue, respectively, with the slope of each fitted line representing the rate of degradation.

experimentally measured mRNA stability changes in cancer cell line models. Specifically, we used time-series measurements of 4-thiouridine-labeled RNA[23] from the MDA-MB-231 cell line, a model of breast cancer, as well as the highly invasive MDA-LM2 cells to identify mRNAs that are differentially stable between these two cell lines (Fig. 2e, see Methods for details; measurements are provided in Supplementary Data 3a). We then compared these experimental stability measurements to RNA-seq-based differential stability estimates between highly metastatic and poorly metastatic PDX models of breast cancer[24–26]. We observed that the mRNAs that are more stable in the invasive MDA-LM2 cell line (based on experimental stability

measurements) are also overall more stable in the highly metastatic PDXs compared to the poorly metastatic PDX (based on DiffRAC analysis of tissue RNA-seq data). Similarly, mRNAs that are less stable in the MDA-LM2 cell line are overall less stable in the poorly metastatic PDX (Fig. 2f; measurements are provided in Supplementary Data 3b).

Interestingly, we found that the mRNAs that are more stable in primary breast tumours compared to normal tissue (based on DiffRAC analysis of TCGA data) are also overall more stable in the highly invasive LM2 line compared to the parental MDA line, and tumour-destabilized mRNAs are overall less stable in the LM2 line (Fig. 2g). This concordance can also be observed at the pathway

level: two of the three pathways that were upregulated in breast tumours based on DiffRAC estimates also appear to be enriched among mRNAs that are stabilized in MDA-LM2 compared to MDA-MB-231 cell lines (MYC targets and mTORC1 signaling, Fig. 2h; example genes are shown in Fig. 2i), supporting a role of mRNA stability in deregulation of these key pathways.

Since the MDA-LM2 line is more invasive than MDA-MB-231, the above analysis suggests that, at least in breast cancer, normal-to-tumour stability changes persist during the progression of the disease to metastasis. To understand whether normal-to-tumour stability changes are correlated with progression-associated stability changes across other cancers, we used DiffRAC to examine the effect of tumour stage and grade on mRNA stability in each TCGA cancer type, by including stage/grade (as numerical variables) in DiffRAC's GLM design while controlling for the confounding effects of age, sex and tumour purity (Supplementary Data 4). The differential stability results therefore reflect the change in stability that occurs as tumour stage or grade increases. We identified a total of 1966 transcripts with significant stability changes associated with tumour stage in at least one of the 11 cancers types that we analysed (Supplementary Data 5a), and 2013 transcripts whose stability was associated with tumour grade in at least one of the four cancer types for which this type of classification was available (Supplementary Data 6). We observed highly cancer-specific associations both for stage and grade (Fig. 3a). Importantly, we found that in most cases the stage- and grade-associated stability changes correlate with normal-to-tumour stability changes (Fig. 3b shows an example, with the overall results summarized in Fig. 3c).

We note that disease progression is often accompanied by substantial cell composition changes, which may confound the estimation of stage/grade-associated stability changes from bulk RNA-seq data. However, previous research has shown that cell type-specific gene expression changes can be identified from bulk RNA-seq data[27]. We implemented a similar design using DiffRAC to deconvolve the stage-associated stability changes occurring specifically in the malignant cells from those occurring in the tumour microenvironment, as well as changes that simply reflect cell composition differences (Fig. 3d, see Methods for details). We identified 275 genes whose stage-associated mRNA stability changes were confidently attributed to dysregulation in malignant cells (Fig. 3e and Supplementary Data 5b). With the exception of one cancer type, the stage-associated stability changes inferred from the tumour bulk were better correlated with the deconvoluted changes attributed to malignant cells compared to those of tumour microenvironment (Fig. 3f, g). Stage-associated changes that could be attributed to malignant cells were also positively correlated with tumour-to-normal changes in most cancer types (Fig. 3h). Taken together, these results highlight widespread mRNA stability changes in tumours, which affect key cancer-related pathways and continue to remodeling of the transcriptome in malignant cells through disease progression.

**RNA-binding proteins play a key role in shaping the tumour mRNA stability profile**. RNA-binding proteins (RBPs) and micro-RNAs (miRNAs) are the key regulators of mRNA stability. These sequence-specific factors primarily affect RNA stability through binding to the 3′ untranslated region (UTR) of their targets–RBPs either stabilize or destabilize their targets[28], while miRNAs primarily destabilize their target mRNAs[29,30]. Starting with RBPs, we set out to examine whether these factors underlie the mRNA stability changes in cancer. We specifically tested for the enrichment of the targets of each RBP among mRNAs that are differentially stable between tumour and normal tissues, after correcting for the background frequency of RBP

binding to each transcript (see Methods). Figure 4a shows an example, where the binding targets of the RBFOX1 protein are enriched among transcripts that are destabilized in glioblastoma multiforme (GBM), relative to the binding targets of other RBPs. We can quantify this enrichment by statistical modeling of the relationship between the binding of a specific RBP to the 3′ UTR of a transcript and the tumour-specific stability status of that transcript (Fig. 4b). We performed a systematic quantification of these relationships for 35 RBPs whose stability target sets (regulons) have been previously mapped based on the presence of their preferred binding sequences in the 3′ UTRs as well as the expression pattern of the candidate target genes[28]. This analysis revealed significantly enriched regulons among tumour-stabilized or destabilized mRNAs across different cancer types, representing deregulation of 17 out of the 35 examined RBPs in at least one cancer type (Fig. 4c). Importantly, we observed excellent agreement between cancer-associated RBP expression changes and RBP target enrichments, after taking into account the expected function of each RBP in stabilizing or destabilizing its targets (Pearson correlation 0.61; Fig. 4d). For example, SNRPA, which is an RNA-destabilizing factor[28], is upregulated in multiple cancers, consistent with the observed destabilization of its regulon (Fig. 4c, d). This strong correlation highlights the reliability of our regulon analysis approach for identifying dysregulated RBPs, and suggests that aberrant expression of RBPs in cancer drives coordinated changes in the stability of their regulons.

Among the RBPs we analysed, two RBPs, namely RBFOX1 and RBFOX3, stand out as being consistently deregulated across several cancer types. Specifically, the targets of these RBPs are enriched among destabilized mRNAs in almost half of all the cancer types we analysed (Fig. 4c). Consistent with the role of RBFOX proteins in promoting mRNA stability[28,31], both RBFOX1 and RBFOX3 are downregulated across multiple cancers (Fig. 5a, b), suggesting that downregulation of RBFOX proteins leads to destabilization of their targets. For both RBFOX1 and RBFOX3, the highest expression in normal tissues can be seen in the brain tissue; subsequently, the most prominent case of their downregulation as well as the most significant changes in the stability of their regulons can be seen in GBM, suggesting a major role in determining tumour transcriptome in this cancer type. However, their effect is not limited to GBM, especially for RBFOX3, which shows a broader range of expression in normal tissues and is also downregulated in a greater number of cancers (Fig. 5b).

To confirm that the downregulation of RBFOX proteins accompanies destabilization of their direct binding targets in cancer, we used HITS-CLIP data of Rbfox proteins in whole brain tissue lysate of mice[32] to build a high-confidence stability network of transcripts that have the strongest binding sites in their 3′ UTRs (see Methods). We confirmed that RBFOX binding sites identified from mouse HITS-CLIP data are conserved in human (Fig. 5c), and observed overall destabilization of the associated targets across different cancers (Fig. 5d). We noticed a subset of mRNAs that are consistently destabilized across the same cancers in which either RBFOX1 or RBFOX3 is downregulated (Fig. 5d). Interestingly, a subgroup of these mRNAs is stabilized in the few cancer types in which RBFOX1 is upregulated (e.g. genes with positive mRNA stability values for LUSC, LUAD and THCA in Fig. 5d), further supporting the notion that their cancer-associated stability changes are driven by RBFOX proteins.

To verify that the stability of these mRNAs is regulated by RBFOX1, we examined the RNA-seq data from differentiated primary human neural progenitor (PHNP) cells in which RBFOX1 is knocked down[33,34]. As expected, cancer-destabilized mRNAs that were associated with RBFOX1 were also downregulated upon RBFOX1 knockdown (Supplementary Data 7a and Fig. 5e). In contrast, when RBFOX1 expression is restored ectopically in mouse neurons lacking RBFOX proteins[31,35], the expression of these genes

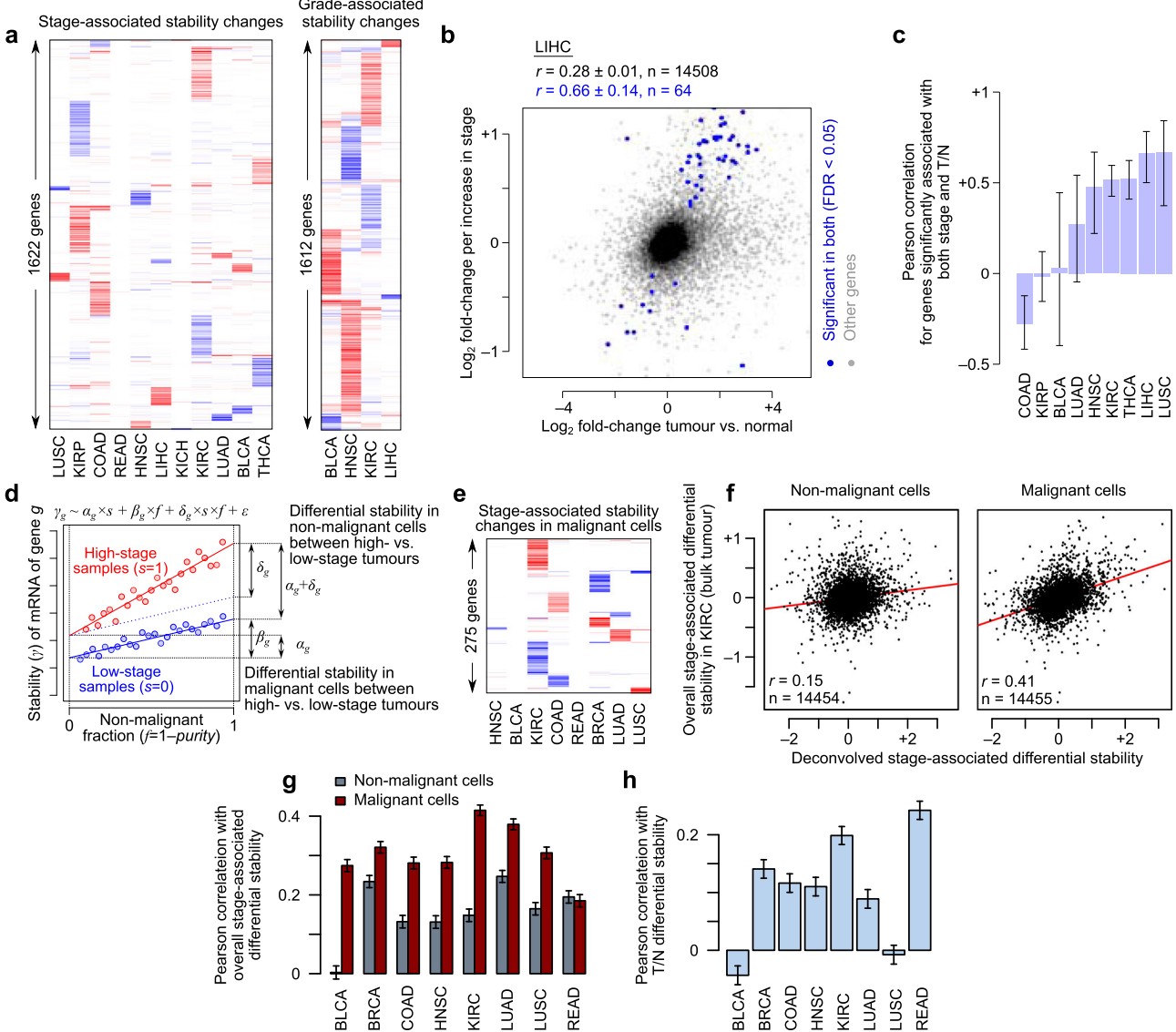

**Fig. 3 Stage- and grade-associated mRNA stability changes. a** The mRNA stability changes associated with tumour stage (left) and grade (right) across TCGA cancers. Genes with significant changes in at least one cancer at FDR < 0.05 are included. The colour gradient is the same as in Fig. 2b. **b** Comparison of the differential mRNA stability between tumour and normal (x-axis) and stage-associated differential mRNA stability (y-axis), in the TCGA-LIHC dataset as an example. Genes with significant changes along both axes at FDR < 0.05 are coloured in blue. Pearson correlation coefficients and confidence intervals for all genes (black) and significant ones (blue) are shown on top. Panel (**c**) summarizes the Pearson correlations for significant genes in other cancer types (error bars represent the confidence intervals). **d** Schematic illustration of the model used for deconvolving the stage-associated changes in malignant and non-malignant cells. The equation on top represents the model used, with the interpretation of model coefficients shown on the plot. See Methods for details. **e** Stability changes associated with tumour stage across TCGA cancers that could be assigned to cancerous/pre-cancerous cells. Genes with significant DiffRAC results in at least one cancer (FDR < 0.05) are included. The colour gradient is the same as in Fig. 2b, with the exception that the log2 fold-change of mRNA stability ranges from −1 to 1 here. **f** Comparison of the stage-associated differential mRNA stability in non-cancerous cells (x-axis, left) or cancerous/pre-cancerous cells (x-axis, right) to the non-deconvoluted estimates (y-axis) in the TCGA-KIRC dataset. Pearson correlation coefficients and p values are shown on the plot. Panel (**g**) shows this Pearson correlations across all cancer types for non-cancerous (gray) and cancerous/pre-cancerous cells (red). Error bars represent the 95% confidence intervals. **h** The Pearson correlation between tumour vs. normal (T/N) differential stability and the deconvoluted stage-associated differential mRNA stability. Only cancer types with at least 5 significant deconvoluted stage-associated genes are shown. Error bars represent the 95% confidence intervals.

is also rescued (Fig. 5f). We identified a core set of eight transcripts that have RBFOX binding site in their 3′ UTRs, are concurrently destabilized across cancers, are inhibited when RBFOX1 is knocked down, and are upregulated when RBFOX1 expression is rescued (Fig. 5g). Interestingly, half of these genes belong to the calcium signaling pathway (based on KEGG pathways[36], Fisher's exact test $P < 10^{-6}$), suggesting that deregulation of RBFOX proteins primarily affects calcium signaling in cancer cells.

Finally, to validate the role of RBFOX1 downregulation in mediating mRNA stability changes in human glioblastoma cells and to investigate whether restoring RBFOX1 activity can rescue the destabilization of its target transcripts, we overexpressed RBFOX1 in the human glioblastoma cell line A172 (Supplementary Fig. 6) and performed RNA-seq. As expected, we observed widespread changes in gene expression (Fig. 5h and Supplementary Data 7b), with overall upregulation of the RBFOX1 regulon

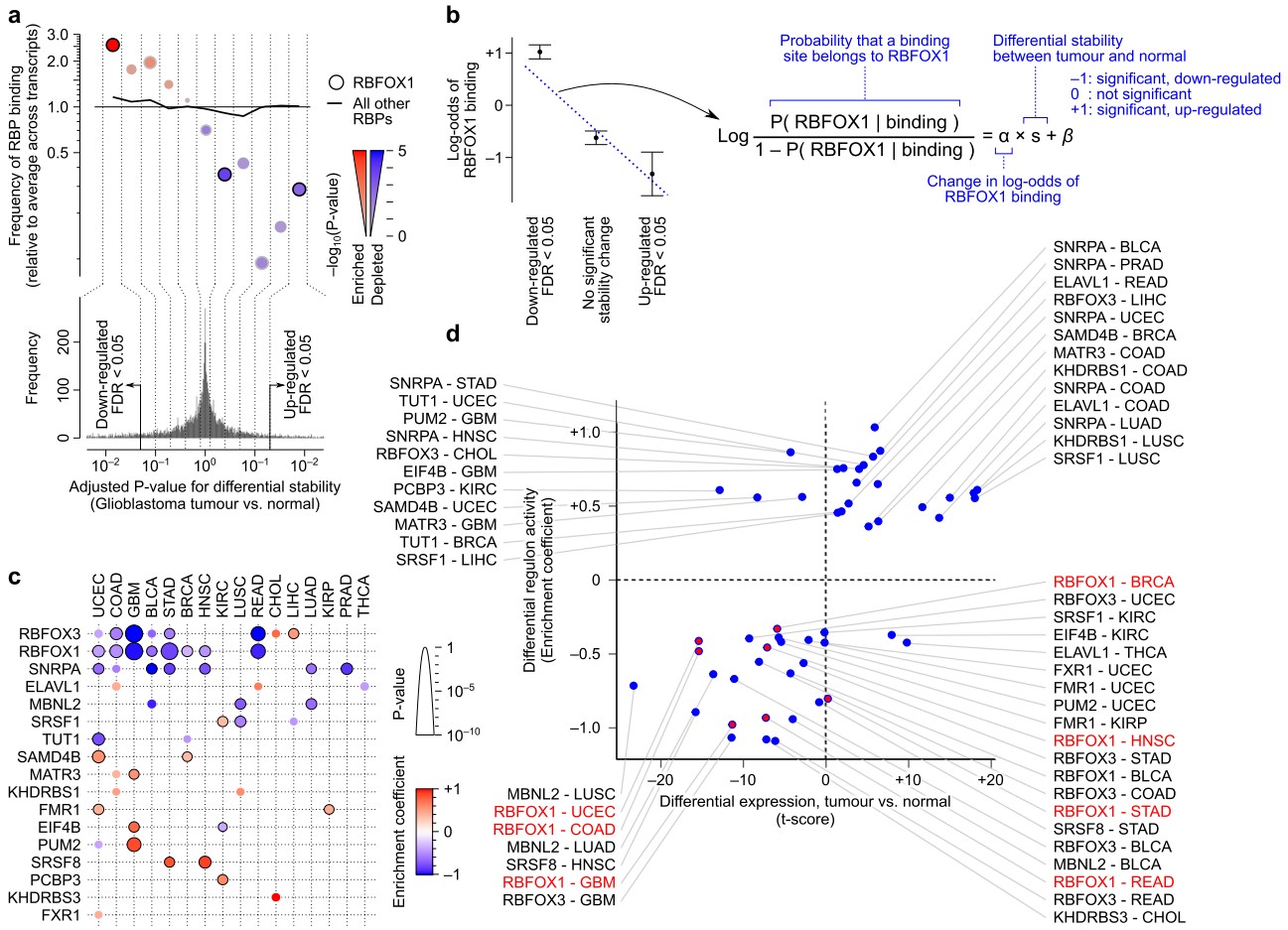

**Fig. 4 Enrichment of RBP binding sites among differentially stabilized mRNAs in cancer. a** An example case showing the enrichment of RBFOX1 binding sites among differentially stabilized mRNAs in TCGA-GBM. Genes are binned by FDR of their DiffRAC differential mRNA stability between tumour and normal, with destabilized mRNAs on the left and stabilized mRNAs on the right. The relative frequency of RBFOX1 targets (circles) and targets of all other RBPs (solid line) is shown for each bin. **b** Schematic representation of the logistic regression approach for modeling the enrichment of RBFOX1 targets (relative to other RBPs) as a function of differential stability. **c** Heatmap summarizing the results of applying the model in panel (**b**) to all RBPs. Positive (red) and negative (blue) regression coefficients indicate enrichment of RBP targets among mRNAs that are stabilized and destabilized in cancer, respectively. The circle size represents the significance level. Significant associations between RBP binding and stability status are shown using black outlines (FDR < 0.05). **d** Comparison of the differential RBP expression (tumour vs normal) and cancer-associated regulon activity. Regulon activity is defined to be the same as the enrichment coefficients from panel (**c**), with the sign of the coefficient inverted for RBPs whose binding leads to RNA destabilization (based on ref. [28]). Each dot represents one RBP in one cancer type. RBFOX1 regulon activities are highlighted. Pearson correlation of differential expression vs. differential regulon activity is 0.61.

in the RBFOX1-overexpressing A172 cell line (Fig. 5i). Consistent with the pathway analysis described above, we observed significant upregulation of calcium signaling pathway genes after RBFOX1 overexpression (Fig. 5j). Furthermore, the majority of pan-cancer destabilized mRNAs that are bound by RBFOX1 are upregulated in A172 cells after RBFOX1 overexpression (Fig. 5k). These results suggest that RBFOX1 downregulation in glioblastoma cells leads to destabilization of its targets, including calcium signaling pathways genes, which can be partially rescued through RBFOX1 overexpression.

**Dysregulation of miRNA regulons shapes the cancer transcriptome.** To examine the contribution of miRNAs to the dysregulation of mRNA stability in cancer, we systematically searched for miRNAs whose targets are disproportionately dysregulated at the stability level in cancer, similar to the RBP analysis above (Methods). Figure 6a shows miR-122 as an example; miR-122 is the most abundant miRNA expressed in liver cells[37], was previously shown to be downregulated in

cholangiocarcinoma, and acts as a tumour suppressor via suppression of cell proliferation and induction of apoptosis[38,39]. As expected, our regulon analysis indicates that miR-122 targets are predominantly stabilized specifically in cholangiocarcinoma tumours compared to normal tissue (Fig. 6a), consistent with reduced activity of miR-122. This observation is consistent with TCGA miRNA expression data, which show specific downregulation of miR-122 expression in cholangiocarcinoma (Supplementary Fig. 7). Systematic application of this network-based approach revealed that, out of 153 broadly conserved miRNA families, the regulons of 63 miRNAs are deregulated in at least one cancer type, suggesting widespread disruption of miRNA networks (Fig. 6b).

Of interest, we observed that miR-29 targets are recurrently stabilized across more than half of the cancer types we analysed, suggesting a pan-cancer decrease in miR-29 activity. Among these cancer types, the miR-29 regulon showed the most significant enrichment among stabilized mRNAs in UCEC and KIRC (clear cell renal cell carcinoma), suggesting a major role in post-transcriptional

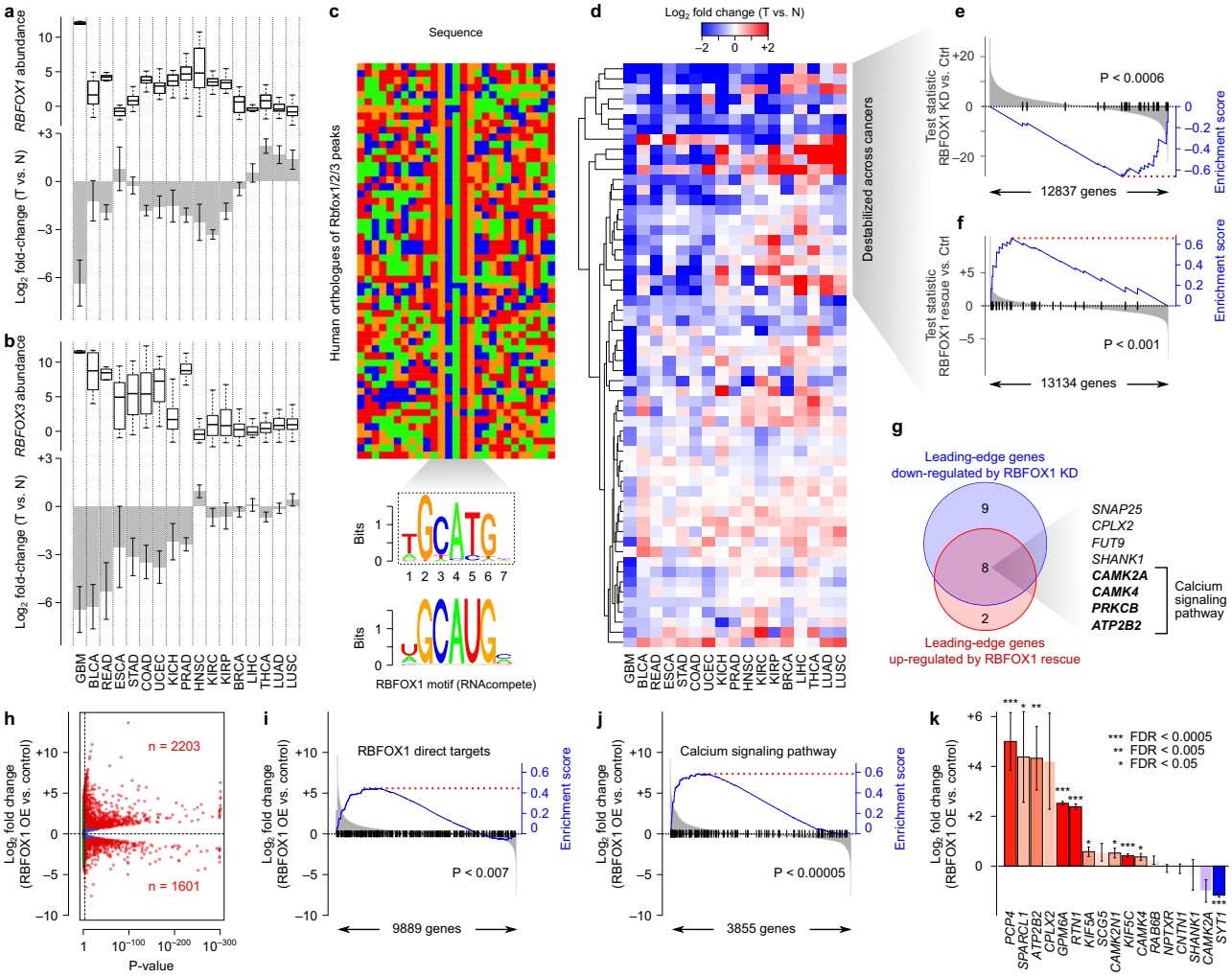

**Fig. 5 Aberrant activity of RBFOX proteins mediates stability changes across multiple cancers. a** *RBFOX1* expression across TCGA cancer types. The box plot (top) shows the *RBFOX1* log$_2$(RSEM) gene expression, retrieved from Firebrowse (http://firebrowse.org/), in normal tissue samples. The bar plot (bottom) illustrates the average log fold-change of *RBFOX1* expression in tumours compared to normal samples (T vs. N; error bars represent SEM). **b** *RBFOX3* expression in normal tissue samples and differential expression in tumours, similar to panel **a**. **c** Conservation of mouse RBFOX1 binding sites in humans for high-confidence Rbfox HITS-CLIP targets[32]. The heatmap shows the sequences of human orthologs of the mouse Rbfox binding sites (Rbfox motif hits on the mouse sequences were identified, and the orthologous regions were extracted using liftOver[69]. The consensus sequence from the human orthologs is shown underneath the heatmap. The RBFOX1 motif from RNAcompete[28] is also shown at the bottom. **d** Heatmap showing the stability of RBFOX HITS-CLIP targets (as defined above). Rows correspond to genes and columns to cancer types, with the latter sorted in the same order as panels (**a**, **b**). **e** Gene set enrichment analysis (GSEA)[70] for RBFOX1 inhibition in terminally differentiated neurons. Genes (x-axis) are sorted by the Wald test statistic of differential expression between RBFOX1 knockdown (KD) and control (Ctrl) cells, with vertical black lines demarcating the pan-cancer-destabilized set of RBFOX1 targets. The blue line represents the enrichment curve for this gene set[70]. **f** GSEA for RBFOX1 rescue in mouse neurons deficient for RBFOX proteins, similar to panel (**e**). **g** Venn diagram illustrating the overlap between the leading-edge[70] set of genes downregulated by RBFOX1 knockdown (from **e**) and the leading-edge set of genes upregulated by RBFOX1 rescue (from **f**). **h** Volcano plot of differential gene expression in RBFOX1-overexpressing (OE) A172 cells. **i** Gene set enrichment analysis (GSEA) (Subramanian et al., 2005) for RBFOX1 overexpression (OE) in the A172 human glioblastoma cell line. Genes (x-axis) are sorted by the log$_2$ fold change of differential mRNA stability between RBFOX1 overexpressing and control cells. The blue line represents the enrichment curve for RBFOX1 direct targets. **j** Similar to panel **i**, with the blue line representing the enrichment curve for genes involved in the calcium signalling pathway. **k** Differential gene expression in RBFOX1-overexpressing A172 cells (n = 3 biological replicates) relative to controls (n = 3 biological replicates), shown for pan-cancer destabilized mRNAs that are bound by RBFOX1 (from panel **d**). Error bars represent the SEM.

remodeling in these cancer types. To understand whether restoring miR-29 activity can reverse these post-transcriptional changes, we expressed a miR-29 mimic in 786-O and A-498 cells, which are models for KIRC (Supplementary Fig. 8). As expected, expression of miR-29 mimic resulted in global downregulation of the miR-29 regulon (Fig. 6c, Supplementary Fig. 9a, and Supplementary Data 8a, b). Importantly, miR-29 mimic expression leads to downregulation of the majority of mRNAs that are significantly stabilized in KIRC (Fig. 6d and Supplementary Fig. 9b), most of which have a miR-29

binding site in their 3′ UTRs. Conversely, miR-29 inhibition in the ACHN cell line (also a model for KIRC) reversed these patterns, with a global upregulation of miR-29 targets (Supplementary Fig. 10 and Supplementary Data 8c), and upregulation of transcripts that are stabilized in KIRC and potentially targeted by miR-29 (Fig. 6e). Together, these results suggest that miR-29 downregulation has a widespread effect on the stability of transcripts in cancer, while restoring its activity partially rescues the normal mRNA stability landscape of the cell.

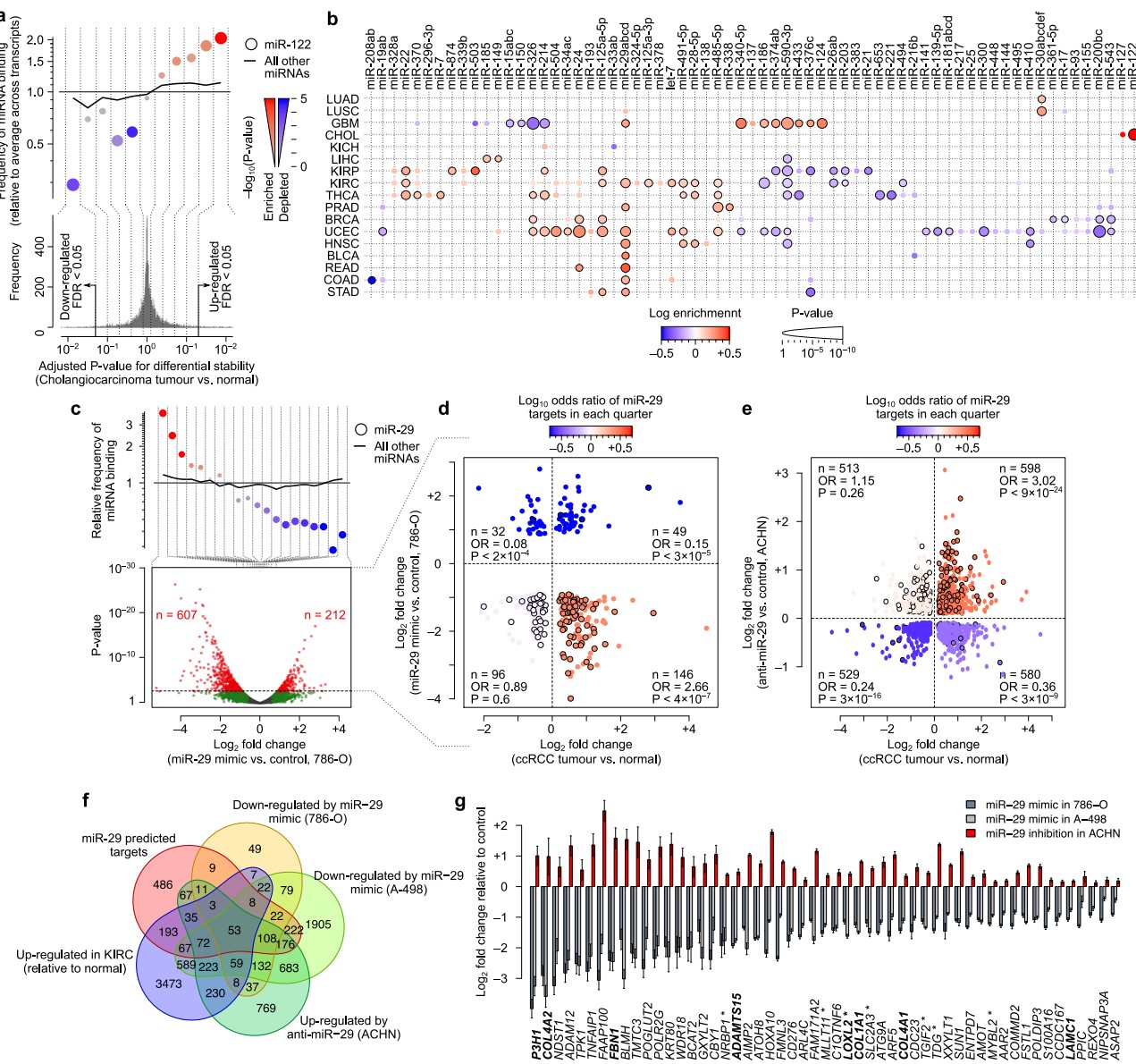

**Fig. 6 Dysregulation of miRNA regulons in cancer. a** An example case showing the enrichment of miR-122 targets among mRNAs stabilized in TCGA-CHOL tumours (relative to normal), similar to Fig. 4a. Note that since miRNAs are expected to destabilize their targets, enrichment in differentially stable mRNAs indicates downregulation of miRNA activity. **b** Heatmap summarizing enrichment analysis for all miRNAs across all cancer types, similar to Fig. 4c. **c** Enrichment of miR-29 targets among genes that are downregulated after transfection of miR-29 mimic in 786-O cells (n = 1) relative to control (n =1). The volcano plot (bottom) summarizes differential expression results between miR-29 mimic and control; the dot plot at the top shows enrichment of miR-29 targets at bins of differentially expressed genes, similar to panel (**a**). In the volcano plot, significantly differentially expressed genes (FDR < 0.05) are shown in red. **d** Enrichment of miR-29 binding sites, relative to other miRNA binding sites, in genes categories defined by their differential mRNA stability in TCGA-KIRC and differential expression after miR-29 mimic expression in 786-O cells. Each dot represents a gene, and those with a black outline contain at least one miR-29 binding site. The colour gradient represents the log-odds of miR-29 binding site enrichment in each quarter. P values are based on Fisher's exact test. Also see Supplementary Fig. 8 for miR-29 mimic expression in 786-O cells. **e** Similar to panel (**d**), but using differential expression after miR-29 inhibition in ACHN cells. **f** Venn diagram illustrating the overlap of genes that are bound by miR-29, upregulated in KIRC, downregulated after miR-29-mimic treatment of 786-O and A-498 cells, and upregulated after miR-29 inhibition in ACHN cells. **g** Differential expression of the 53 genes identified in panel (**f**), in 786-O or A-498 cells expressing a miR-29 mimic, or in ACHN cells expressing a miR-29 inhibitor. Error bars represent the SEM. Genes that are bold correspond to ECM genes (based on overlap with GO), and those with an asterisk are markers of embryonal carcinoma (based on StemCheker (Pinto et al., 2015)).

## Discussion

By quantifying differential mRNA stability patterns across 18 cancer types, our study presents a systematic resource for mining the post-transcriptional landscape of cancer. Importantly, our results uncovered recurrent changes in the stability of >13,000 mRNAs in at least one cancer type, highlighting the widespread

role of post-transcriptional regulation in shaping the cancer transcriptome. We note that this resource also provides an approximation for the relative contribution of transcriptional and post-transcriptional events in shaping cancer transcriptome: on average, 19% of genes that are significantly upregulated at the expression level are detected by DiffRAC as significantly stabilized in tumours,

and 23% of genes with significantly reduced expression are detected as significantly destabilized. In comparison, 66% and 61% of genes whose expression is significantly up- or downregulated are detected as transcriptionally activated or inhibited in tumours, respectively (Supplementary Fig. 11). We note that about 57% of the variability in the number of differentially stabilized genes across cancer types appears to be attributed to sample size, suggesting that our analysis may be underpowered for smaller cancer cohorts (Supplementary Fig. 12). Nonetheless, these results suggest an important role for post-transcriptional changes in shaping the cancer transcriptome, with recurrent changes that are ~30% as frequent as transcriptional events.

Our study also highlights the coordinated post-transcriptional deregulation of genes that are involved in the same pathways. Notably, we observed recurrent stabilization of mRNAs that encode epithelial-mesenchymal transition (EMT) proteins and MYC targets across multiple cancer types. EMT is the process by which epithelial cells lose their apical-basal polarity and cell–cell adhesion, and instead acquire mesenchymal properties such as migratory and invasive potentials[40]; our results suggest that activation of the EMT pathway in cancer is at least partly mediated by post-transcriptional upregulation. Similarly, we observed post-transcriptional upregulation of MYC targets, which include growth-related genes that directly contribute to tumourigenesis[41]. MYC is a well-defined transcription factor and represents one of the most frequently amplified oncogenes[42], leading to transcriptional activation of its targets in cancer. Therefore, our intriguing observation that MYC targets are also upregulated at the mRNA stability level suggests the presence of convergent transcriptional and post-transcriptional mechanisms that modulate overlapping gene sets. Furthermore, we observed coordinated destabilization of mRNAs for genes implicated in oxidative phosphorylation (OXPHOS) and related pathways such as fatty acid metabolism and adipogenesis, consistent with the well-documented Warburg effect in which upregulation of glucose consumption and glycolysis is accompanied by a downregulation of OXPHOS[43].

In addition, we observed widespread and coordinated post-transcriptional modulation of the targets of RNA-binding proteins (RBPs) in cancer, with the RBFOX family of RBPs standing out as having the most recurrently downregulated regulon across multiple cancer types. RBFOX proteins are known regulators of alternative splicing and mRNA stability[28] and have been implicated in a number of neurological diseases[17,31,44], but their role in cancer is less characterized. Nonetheless, at least the *RBFOX1* locus appears to be among the most frequently deleted loci across different cancer types[45,46], with its deletion[47] or other genetic defects[48] being associated with poor survival. Our study suggests that downregulation of RBFOX proteins leads to destabilization of their target transcripts in tumours; many of these transcripts encode proteins involved in calcium signaling, a critical pathway that affects a wide range of cancer-associated processes such as proliferation, invasion, and apoptosis[49]. The association between RBFOX1 and calcium signaling is also supported by previous literature that shows a positive effect of RBFOX1 on the expression of some of the genes involved in this pathway[50]. We note that the RBFOX family of proteins includes RBFOX1, RBFOX2, and RBFOX3; however, *RBFOX1* and *RBFOX3* show the greatest extent of downregulation across different tumours (>60-fold, Fig. 5a, b), whereas *RBFOX2* shows comparatively moderate downregulation (~3-fold, Supplementary Fig. 13). Furthermore, *RBFOX2* does not show significant correlation with the expression of the mRNAs that contain the RBFOX-binding consensus sequence[28]. Taken together, these observations suggest that RBFOX1/3 are the most likely candidates driving dysregulation of the RBFOX regulon in cancer.

In addition to RBPs, our results also highlight cancer type-specific deregulation of mRNA stability by miRNAs, with miR-29 standing out as a pan-cancer stability factor. Our observations are in line with previous studies showing that different miR-29 isoforms act as tumour suppressors and are downregulated in several cancer types[51,52], affecting cell proliferation, differentiation and apoptosis[53]. This downregulation correlates with more aggressive forms of cancer, characterized by increased metastasis, invasion and relapse[54], and therapeutic restoration of miR-29 was suggested to improve disease prognosis[55]. In line with these reports, we observed pan-cancer stabilization of miR-29 targets, suggesting widespread reduction in miR-29 activity in cancer, which could be partially reversed by miR-29 rescue. We note that our results highlight a core set of 53 mRNAs that are miR-29 targets, stabilized at least in KIRC, downregulated after restoring miR-29 activity in the KIRC model cell lines 786-O and A-498, and upregulated after miR-29 inhibition in ACHN cells (Fig. 6f). Importantly, seven of these genes are markers of embryonal carcinoma, suggesting that miR-29 inhibition is essential for activation of an embryonic-like program in cancer (Fig. 6g). In addition, we observed a significant enrichment of the extracellular matrix (ECM) genes (Fig. 6g), suggesting that miR-29 inhibition also contributes to ECM remodeling in cancer, consistent with previous reports on ECM regulation by miR-29[56].

It should be noted that various pathways may affect mRNA stability and its estimates. For example, disruptions in the nonsense-mediated decay (NMD) pathway affects the translation-dependent stability of a wide range of mRNAs[57]. Since most of the affected transcripts are likely spliced[58], such changes are expected to be properly captured by our analysis of spliced/unspliced transcript ratios. However, analysis of spliced/unspliced transcript ratios may not be suitable for studying NMD-dependent clearance of unspliced cytoplasmic transcripts[59]. Other proteins involved in the RNA decay pathway are also expected to influence mRNA stability, although we were not able to detect a significant association between the degree of RNA stability disruption and somatic alterations in RNA decay pathway proteins (Supplementary Fig. 14). While RNA surveillance pathways such as NMD and general RNA decay proteins affect mRNA stability globally, in this work we chose to focus on regulon-specific disruptions caused by abnormal activity of RBPs and miRNAs. We note that different mechanisms may underlie the observed disruption in the RBP/miRNA regulons in cancer, including changes in the expression levels of these regulatory factors, mutations, post-translational modifications in the case of RBPs, disruption of miRNA biogenesis, competition/cooperation with other regulatory factors, and enhanced/restricted access to binding sites on target transcripts. However, at least in the case of RBPs, we observed a strong correlation between their expression and regulon activity in cancer (Fig. 4d), suggesting that disruption of the expression of RBPs is most likely the dominant mechanism underlying the dysregulation of their regulons.

Together, these results highlight a key role for mRNA stability programs, mediated by RBPs and miRNAs, in regulation of pathways that are integral to cancer development and progression. While the vast majority of current literature is focused on the role of transcriptional mechanisms in reprogramming cancer cells, this study underlines a critical and largely uncharacterized role for post-transcriptional remodeling of the cancer cell transcriptome, and provides a resource for exploring post-transcriptional pathways in cancer.

## Methods
**Joint modelling of intronic and exonic read counts and mRNA stability.** Our approach for statistical modeling of intronic and exonic read counts builds on

previous research that connects the abundance of pre-mRNA and mature mRNA to mRNA stability (Supplementary Fig. 1a, b):

$$\log \boldsymbol{m} = b \times \log \boldsymbol{p} + \log \varphi + \log \boldsymbol{\gamma} \tag{1}$$

here, $m$ corresponds to the vector of the mature mRNA abundance for a given gene across different samples, $p$ is the abundance of the pre-mature mRNA, $\gamma$ is the mRNA stability across samples, $\varphi$ is the maximum processing rate of RNA, and $b$ is the bias-term (Supplementary Fig. 1b). Vectors are differentiated from scalars using bold typeface.

We further model the logarithm of mRNA stability as a linear function of a set of sample-level variables:

$$\log \boldsymbol{\gamma} = \boldsymbol{X} \times \boldsymbol{\beta} + \alpha \tag{2}$$

here, $X$ is the $n \times k$ matrix of sample-level variables (for $n$ samples and $k$ variables), $\beta$ is the vector of coefficients that quantify the effect of each variable on the mRNA stability, and $\alpha$ is an intercept (matrices are differentiated from vectors using capital letters). This leads to:

$$\log \boldsymbol{m} = b \times \log \boldsymbol{p} + c + \boldsymbol{X} \times \boldsymbol{\beta} \tag{3}$$

where $c = \log \varphi + \alpha$. We model the mean of intronic read counts for a given gene across samples as a function of the pre-mRNA abundance for that gene, a gene-level scaling factor that can be interpreted as the effective length, and a sample-specific scaling factor that can be interpreted as library size (Fig. 1b):

$$\boldsymbol{\lambda}^{int} = \boldsymbol{p} \times l \times \boldsymbol{s}^{int} \tag{4}$$

here, $int$ stands for intronic, $\lambda$ represents the mean read count, $l$ is the gene-specific scaling factor, and $s$ is the sample-specific scaling factor. Similarly, the mean of exonic read counts for a given gene across samples can be expressed as:

$$\boldsymbol{\lambda}^{exo} = \boldsymbol{m} \times l' \times \boldsymbol{s}^{exo} \tag{5}$$

The above equations can be collectively expressed by matrix operations as:

$$\log \begin{bmatrix} \boldsymbol{\lambda}^{int} \\ \boldsymbol{\lambda}^{exo} \end{bmatrix} = \log \begin{bmatrix} \boldsymbol{s}^{int} \\ \boldsymbol{s}^{exo} \end{bmatrix} + \boldsymbol{X}' \begin{bmatrix} \log \boldsymbol{p}' \\ c' \\ \boldsymbol{\beta} \end{bmatrix} \tag{6}$$

where

$$\boldsymbol{X}' = \begin{bmatrix} \boldsymbol{I}_n & \boldsymbol{0}_{n \times 1} & \boldsymbol{0}_{n \times k} \\ b \times \boldsymbol{I}_n & \boldsymbol{1}_{n \times 1} & \boldsymbol{X}_{n \times k} \end{bmatrix} \tag{7}$$

and $p' = p \times l$, $c' = c + \log(l') - b \times \log(l)$, and $I$ is the identity matrix (matrix dimensions are indicated as subscripts). These equations connect pre-/mature mRNA abundance and mRNA stability to the observed intronic and exonic read counts for each given gene (see Supplementary Fig. 1c, d for matrix equations that consider all genes at the same time). This formulation enables the estimation of unknown parameters using a generalized linear model with a log-link function. In this study, we use DESeq2[60] to fit the unknown parameters of this model, as explained below.

It should be noted that changes in the ratio of spliced/unspliced mRNAs, and ultimately in the observed intronic and exonic read counts, may arise from a wide array of pathways affecting decay of pre-mRNAs or mature mRNAs in different manners. However, previous research has demonstrated that nuclear decay of pre-mRNAs does not affect the ratio of exonic/intronic reads[17] (Supplementary Fig. 1b). This indicates that mechanisms affecting pre-mRNA levels do not lead to a substantial change in the final ratio of spliced/unspliced mRNAs as long as the pre-mRNA remains a potential substrate for the splicing machinery, since a change at the pre-mRNA level leads to an equivalent change at the mature mRNA level and, therefore, does not affect the ratio. The estimates of differential stability generated in this study therefore represent mostly the effect of change in degradation occurring at the mature mRNA levels.

Different RNA selection methods can also affect the intronic read counts. Poly(A)-selected RNA will lead to a lower proportion of intronic reads compared to rRNA-depleted RNA. In the current study, we made use of several poly(A)-selected datasets, including the RNA-seq data from TCGA. However, since all samples in each dataset were analysed using the same method, the estimates are all affected in a similar manner across the sample types and cancer types. We note that poly(A)-selected RNA has previously been shown to produce sufficient intronic reads for stability estimation[15]. In addition, the large number of samples included in this study most likely mitigates any statistical power loss that results from lower amount of intronic reads.

### Estimation of the effect of sample variables on mRNA stability

The above equations allow us to estimate the distribution of latent variables $\log p'$, $c'$, and $\beta$ by fitting the model to observed intronic and exonic read counts. For this purpose, we use the matrix $X'$ as the design matrix in a DESeq2 model. In practice, we replace the first column of $X'$ with an intercept (Fig. 1c), which is an equivalent design matrix and does not change the interpretation of $\beta$, but enables the user to employ a beta prior (if desired) when fitting the DESeq2 model.

In order to be able to construct $X'$, the bias term $b$ needs to be first estimated. We do this by first optimizing $b$ in order to maximize the likelihood of observed intronic

and exonic read counts across all genes in a model that assumes the mRNA stability is a gene-specific constant. Specifically, we use the below design matrix $D$ to fit the model using DESeq2, while varying the value of $b$ in the interval [0,1] to select the $b$ that maximizes the sum of log-likelihood of the data across all genes:

$$\boldsymbol{D}' = \begin{bmatrix} \boldsymbol{I}_n & \boldsymbol{0}_{n \times 1} \\ b \times \boldsymbol{I}_n & \boldsymbol{1}_{n \times 1} \end{bmatrix} \tag{8}$$

we use the 'optimize' function in R to select the optimal value of $b$. Once this optimal value is identified, it is used in the matrix $X'$ (see above), which is then used as the design matrix in DESeq2 to estimate the latent variables, including $\beta$ (i.e. the effect of each variable on stability). This procedure is implemented in DiffRAC (https://github.com/csglab/DiffRAC).

### A modified design to accommodate larger sample sizes

A major limitation of this approach is the considerable increase in computing time with larger sample sizes when DESeq2 is used to fit the model, since the model includes sample-specific latent variables for pre-mRNA abundance. To accommodate these cases, we have also implemented a model that assumes that most of the variance in pre-mRNA abundance can be explained by the experimental variables, instead of including sample-specific latent variables:

$$\log \boldsymbol{p} = \boldsymbol{X} \times \boldsymbol{\omega} + \rho \tag{9}$$

Here, $\omega$ is the vector of coefficients that represent the effect of each variable on the pre-mRNA abundance of a given gene, and $\rho$ is a gene-specific intercept. There, we also have:

$$\log \boldsymbol{m} = b \times (\boldsymbol{X} \times \boldsymbol{\omega} + \rho) + c + \boldsymbol{X} \times \boldsymbol{\beta} \tag{10}$$

This leads to a modified set of matrix equations (Supplementary Fig. 3a–c) that connect intronic/exonic read counts to sample variables:

$$\log \begin{bmatrix} \boldsymbol{\lambda}^{int} \\ \boldsymbol{\lambda}^{exo} \end{bmatrix} = \log \begin{bmatrix} \boldsymbol{s}^{int} \\ \boldsymbol{s}^{exo} \end{bmatrix} + \boldsymbol{X}' \begin{bmatrix} \rho' \\ \boldsymbol{\omega} \\ c' \\ \boldsymbol{\beta} \end{bmatrix} \tag{11}$$

where

$$\boldsymbol{X}' = \begin{bmatrix} \boldsymbol{1}_{n \times 1} & \boldsymbol{X}_{n \times k} & \boldsymbol{0}_{n \times 1} & \boldsymbol{0}_{n \times k} \\ \boldsymbol{1}_{n \times 1} & b \times \boldsymbol{X}_{n \times k} & \boldsymbol{1}_{n \times 1} & \boldsymbol{X}_{n \times k} \end{bmatrix} \tag{12}$$

and $\rho' = \rho + \log l$, and $c' = c + \log(l'/l) + \rho \times (b-1)$. Similar to the previous section, $X'$ can be used as the design matrix for DESeq2 to estimate the latent variables, including $\omega$ and $\beta$.

To construct $X'$, the bias-term $b$ is chosen so that it maximizes the sum of log-likelihood of data across all genes in a model that assumes gene-specific constant stability, i.e. with the below design matrix $D'$:

$$\boldsymbol{D}' = \begin{bmatrix} \boldsymbol{1}_{n \times 1} & \boldsymbol{X}_{n \times k} & \boldsymbol{0}_{n \times 1} \\ \boldsymbol{1}_{n \times 1} & b \times \boldsymbol{X}_{n \times k} & \boldsymbol{1}_{n \times 1} \end{bmatrix} \tag{13}$$

This simplified model is also implemented in DiffRAC. Overall, we see strong agreement between DiffRAC's estimates when using the two different models (i.e. sample-specific pre-mRNA abundances vs. condition-specific pre-mRNA abundances) on the same data (Supplementary Fig. 3d).

### Differential RNA stability between NAT10 knockout and parental cells

Raw BRIC sequencing (BRIC-seq) (5'-bromo-uridine [BrU] immunoprecipitation chase-deep sequencing analysis) reads for time-series measurements of BrU-pulsed RNAs in parental and $NAT10^{-/-}$ HeLa cells[20,61] were obtained from GEO accession GSE102113 (SRA accession SRP114504). This RNA-seq dataset represents time points 0, 2, 4, 8 and 16 h after a 24-hour treatment of cells with BrU (two replicates for each cell line at each time point). Reads were mapped to the GRCh38 genome assembly using HISAT2[62], and gene-level read counts for each sample were obtained using HTSeq-count[63] ("intersection-strict" mode) based on Ensembl GRCh38 v87 gene annotations. Ground-truth Differential mRNA stability between the control and NAT10KO cells was obtained using DESeq2[60] by modeling the RNA abundances as a function of $\sim c + t + c{:}t$, where $c$ is the cell type (0 for Control and 1 for NAT10KO), $t$ is the time point, and $c{:}t$ is the interaction between cell type and time. In this model, the coefficient of $c$ would represent the differential expression between the two cell types (i.e. difference in abundance at time zero); the coefficient of $t$ would represent the stability of each gene's mRNA in the reference cell line (relative to the average of all genes); and the coefficient of the interaction term $c{:}t$ would represent the differential mRNA stability between the two cell lines. For each gene, the coefficient of $c{:}t$ and associated statistics were retrieved using DESeq2.

### TCGA RNA-seq data processing

RNA-seq BAM files for 7078 tumour samples and 682 adjacent normal samples from the 18 cancer types with at least 5 normal samples in TCGA were acquired from the National Cancer Institute (NCI) Genomic Data Commons (GDC) data portal (https://portal.gdc.cancer.gov/GDC; dbGaP study accession phs000178.v1.p1). All TCGA RNA-seq data used in this

study was generated from poly(A)-selected RNA. In order to quantify the number of reads corresponding to pre-mRNA and mature mRNA for the estimation of mRNA stability, we generated custom annotations for exons and introns for the transcripts supported by both Ensembl and Havana consortia, using GTF formatted annotations acquired from Ensembl GRCh38 version 87.

We note that, in addition to mRNA stability, aberrant alternative splicing may affect the exonic read profiles. To avoid the potential confounding effect of alternative splicing on mature mRNA quantification, we exclusively retained exonic reads mapping to constitutive exons that are present in all Ensembl/Havana transcripts. Even when only constitutive exons are used for read counting, there might be cases where a splicing shift leads to transcripts that have reduced or enhanced stability. In such cases, DiffRAC should still detect the overall change in stability, even though it is caused by the interaction between abnormal alternative splicing and isoform-specific decay mechanisms. Similar to ref. [17], we limited our analysis of RBP and miRNA regulons to the genes that shared the same 3′ UTR across all their isoforms, with the 3′ UTR composed of a single exon, to mitigate the potential confounding effect of alternative 3′ UTR usage/splicing on mRNA stability.

Intronic regions were included in our annotations only if they did not overlap with any exon, regardless of whether the exon was concordantly annotated by Ensembl or Havana consortia. The strandedness of RNA-seq data was determined using RSeQC[64]. Subsequently, BAM files were sorted by read name using SAMtools, and exonic and intronic reads were separately counted using HTSeq-count[63], limiting to reads with a MAPQ score ≥30. Exonic reads were counted using the HTSeq "intersection-strict" mode, whereas intronic reads were counted using the "union" mode. The exonic/intronic read counts were then used as input to DiffRAC for stability analysis. We removed the cell cycle genes (based on GO term GO:000704) for downstream analyses, given that these genes are not at steady state, which is required for estimating stability from pre-/mature mRNA abundances.

**Deconvolution of cellular origin from differential stability estimates**. We inferred stage-associated changes in stability specifically originating from the cancerous (or pre-cancerous) cells using DiffRAC with a design matrix that models the exonic/intronic read ratio as a function of the tumour stage (dichotomized into low-stage and high-stage categories), the impurity (fraction of non-malignant cells) of the tumour as measured by ABSOLUTE[65], and an interaction term between stage and impurity, similar to ref. [27]. As shown in Fig. 3d, different coefficients retrieved from this model represent the stage-associated changes in stability originating from cancerous or pre-cancerous cells specifically. Specifically, the coefficient of the tumour stage variable represents difference in stability between high- and low-stage tumours when impurity is zero, and thus can be interpreted as the stage-associated differential stability that is confidently attributed to malignant cells.

**Pathway analysis**. MSigDB hallmark gene-sets[66] were retrieved using the msigdbr R package (https://cran.r-project.org/web/packages/msigdbr/index.html). For each TCGA cancer type, Fisher's exact test was used to examine the association between each pathway and the sets of significantly stabilized or destabilized mRNAs, separately.

**Differential RNA stability between MDA-MB-231 and MDA-LM2 cells**. Raw RNA-seq reads for time-series measurements of 4-thiouridine (4sU)-labeled RNA[23,67] from MDA-MB-231 and MDA-LM2 cells were obtained from GEO accession GSE49608 (SRA accession SRP028570). This RNA-seq dataset represents time points 0, 2, 4, and 7 h after a 2-hour treatment of cells with 4sU (four replicates for each cell line at each time point). Raw data was processed and differential mRNA stability between the MDA-MB-231 and MDA-LM2 cells was obtained in the same way as the NAT10KO BRIC-seq data (see above Methods).

**RBP and miRNA regulon analysis**. The stability regulons of 35 RBPs (i.e. the set of mRNAs bound and regulated by each RBP) were obtained from a previous publication[28]. The regulons of miRNA families were obtained by identifying exact miRNA seed matches in mRNA 3′ UTRs. Specifically, 3′ UTR sequences of protein-coding genes were retrieved using the Ensembl GRCh38 version 87 annotations. We limited the analysis to the genes for which a single 3′ UTR, composed of a single exon, was shared across all isoforms, in order to avoid the possible confounding effects of alternative splicing. The miRNA seed sequences (8nt) were retrieved from TargetScan v7.2[68], limiting to a set of 153 broadly conserved miRNA families (family conservation score ≥1). Exact seed sequence matches in 3′ UTR sequences were identified while limiting the search space to a maximum of 2000 nt downstream of the stop codon.

The regulon enrichment among upregulated or downregulated genes was quantified using a logistic regression approach. Specifically, for each cancer type, we modeled the likelihood of being bound by each RBP/miRNA as a function of status, with −1 corresponding to significantly destabilized mRNAs (FDR ≤ 0.05), +1 corresponding to significantly stabilized mRNAs, and 0 corresponding to non-significant mRNAs. To account for the confounding factors that generally affect the

number of binding sites of RNA-binding factors (rather than a specific RBP or miRNA; e.g. 3′ UTR length), we used the total number of binding sites of each mRNA for RBPs or miRNAs as the background. Specifically, we used a generalized linear model of the binomial family, in which the presence of a binding site for the specific RBP or miRNA of interest is considered as "success", and the presence of binding sites for other RBPs or miRNAs considered as "failures". These success/failure counts were modeled as a function of the stability status of the transcript using the glm function in R.

**HITS-CLIP data analysis**. Pooled HITS-CLIP peaks of RBFOX1/2/3 proteins in whole brain tissue lysate of mice were retrieved from a previous study[32]. Peaks occurring in the 3′ UTR with a height greater or equal to 200 overlapping CLIP tags were retained (peak height was extracted from Supplementary Table 1 of the source publication). The mRNAs that had at least one 3′ UTR high-confidence peak were considered high-confidence RBFOX targets, which were further filtered to include only those whose orthologs had expression measurements in TCGA. This resulted in 58 genes, 54 of which also have a 3′ UTR RBFOX binding site based on CIMS analysis of CLIP data.

**Cell culture and transient transfection of miRNA mimics and inhibitors**. The established renal cancer cell line 786-O, A-498 and ACHN as well as the glioblastoma cell line A172 were purchased from the American Type Culture Collection (ATCC; Rockville, MD, USA) and cultured in Dulbecco's Modified Eagle Medium (DMEM) supplemented with 10% fetal bovine serum (FBS) and 1% penicillin/streptomycin (Life technologies) at 37 °C with 5% CO2. For transient transfection, 786-O and A-498 cells (100,000 cells/well in 6-well plates) were reverse-transfected in antibiotic-free medium with 10 nM of miRNA-29 mimic (stem-loop sequence: UGGUUUCGUAUUGGGUGCAUAGAAGUAUUAAUUU UGUAACUUGUCUAGCACCAUUUGAAACCAGU (two biological replicates for A-498, and one for 786-O), mature miRNA sequence: UAGCACCAUUUGAA ACCAGU, ThermoFisher, 4464066) or control mimic (ThermoFisher, 4464058) (two biological replicates for A-498, and one for 786-O) using Lipofectamine RNAiMAX Reagent (ThermoFisher,13778075) according to the manufacturer's recommendations. ACHN cells were transfected either with miR-29 inhibitor (ThermoFisher, 4464084, Assay ID: MH10103) or negative control (ThermoFisher 4464076) using the same protocol described above, with three biological replicates each. Two additional RNA-seq samples related to the miR-29 mimic experiment performed in A-498 cells were excluded due to potential mislabeling of the samples.

**RNA isolation and qRT-PCR analysis of miRNAs**. Total RNA was extracted using All Prep DNA/RNA/miRNA Universal kit (Qiagen) 48 h after transient transfection. RT-PCR was done using TaqMan MicroRNA reverse transcription kit (Applied Biosystems, 4366596). The LightCycler 480 instrument (Roche) was used to perform qRT-PCR analysis of miR-29 and miR-26 using TaqMan Fast Advanced miRNA Assays (ThermoFisher, 4444557) following guidelines provided by the manufacturer. Expression was reported as Ct values (Supplementary Fig. 8).

**Stable cells expressing RBFOX1**. To generate stable A172 cell lines, HEK293T cells were transfected with lentiviral packaging plasmids (psPAX2 and MD2.g) together with a lentiviral expression plasmid for either GFP or RBFOX1 (three biological replicates each) using Lipofectamine 3000. Plasmids pLX317-GFP and pLX317-RBFOX1 were obtained from the TRC3 ORF collection from Sigma provided by McGill Platform for Cellular Perturbation (MPCP) at McGill University. After 48 h, media containing lentiviral particles were collected, filtered through a 0.45 μm syringe filter, and immediately added to A172 cells with 8 μg/ml polybrene. Over-expression of GFP and RBFOX1 were confirmed by fluorescence microscopy (for GFP) or qPCR (for RBFOX1). Total RNA was extracted using the All Prep DNA/RNA/miRNA Universal kit (Qiagen).

**RNA-sequencing and analysis**. Library preparation from total RNA was performed using NEB rRNA-depleted (HMR) stranded library preparation kit according to manufacturer's instructions, and sequenced using Illumina NovaSeq 6000 (100 bp paired-end). RNA-seq reads were aligned to the GRCh38 genome assembly using HISAT2[62], and gene-level read counts were obtained using HTSeq-count[63] ("intersection-strict" mode) based on Ensembl GRCh38 v87 gene annotations. DESeq2[60] was used to compute differential gene expression.

**Statistics and reproducibility**. All statistical analysis were performed using by Bioconductor packages in R (version 4.1.2). The specific statistical tests used for each analysis and the associated measures of statistical significance are indicated within the main text, methods, in the figure, or in their legends. Statistical significance was set at $P < 0.05$ for all analyses and multiple testing correction was performed when applicable using the FDR method. Sample size for TCGA cohort analysis depended on publicly available data. No statistical analysis was performed to select the sample sizes for RNA-seq experiments. To ensure reproducibility for RNA-seq experiments, biological replicates were used and/or the findings were replicated in other cell lines.

**Reporting summary**. Further information on research design is available in the Nature Research Reporting Summary linked to this article.

## Data availability

Data generated during this study are included in this published article and its supplementary files. Additional data and analysis files are available at http://csg.lab.mcgill.ca/sup/pancancer_stability/ and/or via Zenodo (doi:10.5281/zenodo.4404547). RNA-seq data from the miR-29 mimic and inhibitor expression experiments are available via GEO under accession GSE145088. RNA-seq data from the RBFOX1 overexpression experiment are also available via GEO under accession GSE201639. The results published here are in part based on data generated by the TCGA Research Network: https://www.cancer.gov/tcga. Other data used in this paper are available via their source publications as indicated in the article.

## Code availability

DiffRAC is available via GitHub at https://github.com/csglab/DiffRAC.

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

## Acknowledgements

This work was supported by funds from Canadian Institutes of Health Research (PJT-155966), and resource allocations from Compute Canada to H.S.N. H.S.N holds a Canada Research Chair funded by the Canadian Institutes of Health Research. G.P. and R.A. are supported by training scholarships from the Canadian Institutes of Health Research, the Fonds de recherche du Québec–Santé (FRQS), and Oncopole. T.L. has been supported by a Vanier Canada Graduate Scholarship and a training scholarship from the FRQS. Y.R. is a research scholar of the FRQS. The results published here are in part based on data generated by the TCGA Research Network: https://www.cancer.gov/tcga. Lentiviral ORF expression plasmids were provided by the McGill Platform for Cellular Perturbation (MPCP). We thank Dr. Janusz Rak for providing the A172 cell line.

## Author contributions

G.P. and H.S.N. conceived the study, developed the computational methods, analysed the data, and wrote the manuscript. P.J., E.M., T.N., and M.R. performed the miRNA inhibition/mimic and RBP overexpression experiments. R.A. contributed to data processing. T.L. contributed to deconvolution analyses. Y.R. contributed to experimental design and data interpretation. H.S.N. directed the study.

## Competing interests

The authors declare no competing interests.
