## [Peer Review File · Communications Biology]

Reviewers' comments:

Reviewer #1 (Remarks to the Author):

In this manuscript, the authors attempted to infer mRNA decay rate using a public cancer RNA seq dataset. The study is based on the assumption that the change of post-transcriptional control, such as RNA decay, should change the ratio between the un-spliced forms and the spliced forms. The authors constructed a statistical framework to de-couple the mRNA degradation/stabilization factors from the transcriptional induction/repression factors. To evaluate the performance of the newly developed framework, which they named DiffRac, the authors first conducted a series of validation analyses using the data collected from in vitro cell lines. They found that the DiffRac can infer the mRNA turn-over rates, which is consistent with those measured by the actinomycin D system. Then, they applied this method to the RNA seq dataset of TCGA, which includes 7760 samples spanning 18 cancer types. They found that the mRNA stability changes are a frequent event (happening in total of 30% of the cases). More than 80 possible responsible RNA binding proteins were identified. Among them, they particularly focused on RBFOX1 and confirmed its biological roles, using the cell line. For another aspect of the regulation, they also successfully identified a group of responsible miRNAs. Particularly focusing on miR-29, an in vitro analysis allowed the authors to confirm its role for the mRNA stability control. Overall, this is a unique study, exploring novel research filed of the mRNA stability in vivo cancers. The provided data is comprehensive and the presented results are supported by solid experimental/bioinformatic evidences.

Major points:

1. For the cancerous disruption for the mRNA quality control, would a different category of the regulation, such as the NMD/UPF1 pathway, disturb the ratio of spliced/un-spliced forms similarly? It would accumulate the un-spliced form rather than changing the mature spliced form.
2. Is not there any association between the degree of the disruption of the mRNA stability control and the clinical appearances of the cancers, such as the stage or prognosis of the patients?
3. I wonder in how many cases a genomic mutation resided in a RNA binding protein or an miRNA or their target sites were found to be responsible for the observed disruption as the easiest case?

Minor points:

4. It would be helpful if the IGV view of a typical case is presented, which represent how the tag distribution changes and where the binding site of RNA binding protein is located.
5. I wonder if there are any interpretable cases for which DiffRac failed to give a precise inference? For example, occurrence of cancerous alternative splicing, which may trigger a cancer immune response against the aberrant protein may make an influence?
6. Is not there any association of the changes in the RNA stability and the pattern of the cancer driver mutations? Disruption of some pathways may be prone to invoke errors of post-transcriptional controls.
7. As far as I understand, the RNA data used for the input is a mixture of various cell types in cancers. To ensure observed phenomena is the change occurring in cancer or pre-cancerous epithelial cells, it is important to also refer to the information of the surrounding normal cells. Such a "deconvolution" analysis may be possible also utilizing single cell catalog datasets. Also, validation analyses for the deconvolution results on some representative cell types would be also useful.

Reviewer #2 (Remarks to the Author):

Key results

The authors developed and employ a statistical framework, DiffRAC, to assess the contribution of RNA stability to the transcriptome composition of 18 cancer types from RNAseq data deposited in TCGA. They then benchmark these computationally inferred changes in stability by using a previously generated benchmarking dataset for RNA stability. The authors then examine the RNA stability differences in a pair of highly and poorly metastatic breast cancer cells lines using DiffRAC and compare these results to previously generated experimental stability data from 4-thiouridine pulse-chase experiments in these same cell lines. The authors observe a concordance between DiffRAC-derived and experimentally derived changes in transcript stability, and also observe these changes in normal vs. breast tumor samples. More broadly, the authors also observe cancer-type specific associations between transcript stability and tumor stage and/or grade. Next, the authors examine the role of RNA binding proteins in regulating these stability changes. They find associations between different RBPs and their binding motif-based regulons that are cancer-type specific. They then more closely examine the RBFOX proteins and their role in regulating RNA stability in GBM. Using several previously generated datasets, the authors conclude that RBFOX1 and 3 regulate RNA stability in GBM and specifically regulate calcium signaling via stabilization of transcripts. Finally, the authors examine the role miRNAs may play in modulating the observed changes in RNA stability. They find widespread changes in the stability of miRNA regulons across cancer types. They then examine the role of miR-29 in regulating RNA stability in KIRC, and perform a miR-29 gain-of-function experiment in one KIRC cells line, finding that this does result in destabilization of the miR-29 regulon.

General comments

This paper provides a valuable resource for examining RNA stability changes in human cancer samples for which experimental stability measurements cannot be obtained. However, to support the hypotheses based on their computationally inferred stability data, the authors rely heavily on analysis of experimental data generated by other groups, some of which has limited relevance to their hypotheses, and does not provide entirely novel results (ie, the role of miR-29 in cancer is well-documented, see review Kwon et al 2018). It would strengthen their study to provide additional experimental data to support these claims, especially with regard to the role of RBFOX proteins regulating transcript stability. The DiffRAC analysis itself is novel and will be of interest to the community, but the authors must be careful to not overstate the conclusions generated by their fairly cursory analysis of the downstream regulators of RNA stability across cancers.

Major points

1. It is not entirely clear which benchmarking data the authors used for testing DiffRAC. The paper the authors reference contains many datasets, but the authors only show the results from one: mouse ESCs vs differentiated neurons. Please clarify which data was used for benchmarking, and if it was the single dataset please reference the original paper where this data was generated. It is also unclear as to what the authors mean by "ground-truth transcript half-life measurements", is this referring to experimentally generated half-life measurements? On a technical note, if these measurements were made using actinomycin D treatment, as I infer from the text, this is a less accurate method for measuring transcript stability than many of the newer metabolic labeling methods. It may be more useful to benchmark their framework against data generated using a more accurate method of measuring transcript stability. In either case, the method used for measuring transcript stability in their benchmarking data needs to be clearly stated in the text and the figure legend.
2. Please specify the RNAseq method(s) that were used for sequencing the samples in TCGA. How many datasets came from poly(A) selected RNA vs rRNA depleted RNA? If any of the sequencing data was from poly(A) selected RNA are all of the normal vs tumor samples subjected to the same RNA selection method? I assume that the two methods would result in different proportions of intronic vs exonic reads, therefore affecting DiffRAC output.
3. The authors find that MDA-LM2 vs MDA-MB-231 stability changes are correlated with those in BRCA

vs normal tissue in the TCGA data, with some overlap in the transcript sets. However, since the MDA-LM2 and MDA-MB-231 comparison is a model for metastasis (the MDA-MB-231 line is already considered a metastatic cell line) it would be more informative to look at the correlation between measured differential transcript stability in the MDA-LM2 vs MDA-MB-231 cells with stability changes across breast tumor stage and/or grade. Moreover it would be interesting to look at this by breast cancer subtype as the MDA-MB-231 line is a triple negative line, and stability changes may be specific to this subtype.

4. The authors find stage and grade-associated changes in RNA stability according to tumor type. Please explain how this analysis was done- for cancer types with multiple stages or grades do the RNA stability change associations reflect a consistent increase or decrease in stability across tumor stage or grade (ie, least stable in stage 1, intermediate stability in stage 2 and highest stability in stage 3)? Or is it rather stability changes between any of the stages or grades?

5. The authors' case for RBFOX proteins playing a role in RNA stability in GBM, while intriguing, is limited by their reliance on previously generated data. Some of the data they analyze to support their hypothesis comes from experiments on systems different than the one they're making their claim about. Specifically, to generate their "high-confidence stability network" the authors use Rbfox CLIP data collected from whole mouse brain tissue. This is good quality CLIP data, but human and mouse transcriptomes are not identical, and this limitation should be addressed in the text. To make a more convincing claim that this CLIP data mimics RBFOX binding in human neuronal tissue the authors could examine the conservation between the 3'UTRs bound by Rbfox in mouse with their human homologs.

6. I could not find details how the authors' generated their Rbfox high-confidence stability network besides that it was based on a specific CLIP data set and involved 3'UTR binding with height greater than 200. Please specify a unit for 200, and also consider using the CIMS-called peaks from this CLIP data set as this indicates more robust binding than simple number of reads. Also specify data from which Rbfox CLIP experiments was used. The paper referenced performed CLIP on Rbfox1,2 and 3. Please elaborate on this in the methods or in the text.

7. The authors present their analyses of loss-of-function data from differentiated primary human neural progenitor cells with RBFOX1 knockdown as well as gain-of-function data from mouse neurons with RBFOX1 knockout in which RBFOX1 expression has been restored. In both cases the authors are examining the function of RBFOX1 in non-transformed neuronal models (and in one case, again, a mouse model); however, their hypothesis points to a role for RBFOX1 in human tumorigenesis. Therefore, stronger experimental evidence is needed to make a claim about the role of RBFOX proteins role in regulation of transcript stability in human cancers. Ideally this would include overexpression of RBFOX1/3 proteins in human glioblastoma cells lines followed by RNA-seq for DiffRAC analysis, as well as experimental transcript stability measurements in these cell lines. Similar loss-of-function experiments in human glioblastoma cells lines would likewise make their conclusions much stronger. With the data presented I would also caution the authors to be careful about over-interpreting any GO terms enriched in the overlapping genes from the datasets they've analyzed, as they combine data from a variety of non-cancer and non-human experimental systems. Unless they provide further experimentally based evidence (from human glioblastoma cell lines) the authors do not convincingly make their claim about the role of the RBFOX regulon in calcium signaling in human cancers.

8. The authors provide basic experimental evidence that miR-29 is sufficient to downregulate transcripts that harbor miR-29 target sites in one KIRC cell line, and that there is significant overlap between these transcripts and those downregulated in the TCGA KIRC tumor vs normal data. Without further experimental evidence though the authors should not claim that miR-29 downregulation has a widespread effect on the stability of genes in cancer. If the authors would like to support this claim,

they should perform miR-29 overexpression and inhibition experiments across multiple human KIRC cell lines, or across multiple lines from cancers in which the authors found particularly robust evidence for miR-29 regulation.

Minor points

1. Many times throughout the text the authors refer to stabilized genes. To avoid confusion please refer to stabilized transcripts or mRNA when referencing RNA.
2. Related to figure S3, is the variation across cancer types in the number of transcripts with stability changes passing the statistical threshold related to the number of samples analyzed per cancer type or is there another explanation for this variability? Also, I believe a number has been misreported in the COAD graph in this figure, which should be fixed.
3. Is figure 2f simply a graph of data from Goodarzi et al? This figure seems superfluous.
4. In figure 3b please choose colors with better contrast to better visualize the datapoints.
5. Please provide an explanation for identifying RBFOX1 and RBFOX3 as regulators of the observed effects on RNA stability but excluding RBFOX2. All the RBFOX protein are highly similar, and also share the same binding motif. Furthermore, there is a fairly robust RBFOX2 CLIP dataset from the Yeo lab's eCLIP study, which may be complementary to the CLIP data the author's have used, as it is from human cell lines.
7. In figure 4d it may be helpful to highlight the datapoints representing any of the RBFOX related points, as this is a visually busy graph and RBFOX is the regulon the authors focus on.
8. The miR-29 mimetic and control RNA sequences used for the transfection experiment should be reported.
9. In figure S5 it would be helpful to also include a graph showing the relative quantitation values of miR-29 expression relative to the control, along with the graph of the raw Ct data. This will make it much easier to look at relative fold-change in expression of the miRNA.

Please find below our responses to reviewers' comments. Our responses are in blue. The main modifications are also highlighted in the manuscript file.

Reviewer #1 (Remarks to the Author):

In this manuscript, the authors attempted to infer mRNA decay rate using a public cancer RNA seq dataset. The study is based on the assumption that the change of post-transcriptional control, such as RNA decay, should change the ratio between the un-spliced forms and the spliced forms. The authors constructed a statistical framework to de-couple the mRNA degradation/stabilization factors from the transcriptional induction/repression factors. To evaluate the performance of the newly developed framework, which they named DiffRac, the authors first conducted a series of validation analyses using the data collected from in vitro cell lines. They found that the DiffRac can infer the mRNA turn-over rates, which is consistent with those measured by the actinomycin D system. Then, they applied this method to the RNA seq dataset of TCGA, which includes 7760 samples spanning 18 cancer types. They found that the mRNA stability changes are a frequent event (happening in total of 30% of the cases). More than 80 possible responsible RNA binding proteins were identified. Among them, they particularly focused on RBFOX1 and confirmed its biological roles, using the cell line. For another aspect of the regulation, they also successfully identified a group of responsible miRNAs. Particularly focusing on miR-29, an in vitro analysis allowed the authors to confirm its role for the mRNA stability control. Overall, this is a unique study, exploring novel research filed of the mRNA stability in vivo cancers. The provided data is comprehensive and the presented results are supported by solid experimental/bioinformatic evidences.

We thank the reviewer for positive assessment of the study and its novelty.

Major points:

1. For the cancerous disruption for the mRNA quality control, would a different category of the regulation, such as the NMD/UPF1 pathway, disturb the ratio of spliced/un-spliced forms similarly? It would accumulate the un-spliced form rather than changing the mature spliced form.

We have now added a discussion of other mechanisms that may disturb the ratio of spliced/unspliced forms. Specifically, on page 16, we mention the role of NMD/UPF1 pathway in determining the stability of transcripts. We note that since many targets of NMD pathway are spliced transcripts with premature termination codons (Clark et al., Science 2002, PMID 11988574), disruptions in the NMD pathway likely results in the stabilization and accumulation of spliced transcripts in most cases, leading to an increase in spliced/unspliced ratio. However, as the reviewer has noted, NMD disruption may also lead to accumulation of unspliced or insufficiently spliced transcripts (Sayani et al., Mol Cell 2008, PMID 18691968); this is now

mentioned in the manuscript to ensure the reader is aware of the limitations of spliced/unspliced ratio analysis:

Page 16: *“It should be noted that various pathways may affect mRNA stability and its estimates. For example, disruptions in the nonsense-mediated decay (NMD) pathway affects the translation-dependent stability of a wide range of mRNAs (Kurosaki et al., 2019). Since most of the affected transcripts are likely spliced (Clark et al., 2002), such changes are expected to be properly captured by our analysis of spliced/unspliced transcript ratios. However, analysis of spliced/unspliced transcript ratios may not be suitable for studying NMD-dependent clearance of unspliced cytoplasmic transcripts (Sayani et al., 2008).”*

We have also noted in the Methods section that, for pre-mRNAs that are substrates of the splicing machinery (which excludes cytoplasmic unspliced mRNAs such as those studied by Sayani et al. 2008), changes in their decay rate (e.g. through nuclear RNA exosome) does not affect the spliced/unspliced ratio, as previously described by Alkallas et al. (2017):

Pages 19-20: *“It should be noted that changes in the ratio of spliced/un-spliced mRNAs, and ultimately in the observed intronic and exonic read counts, may arise from a wide array of pathways affecting decay of pre-mRNAs or mature mRNAs in different manners. However, previous research has demonstrated that nuclear decay of pre-mRNAs does not affect the ratio of exonic/intronic reads (Alkallas et al., 2017) (Supplementary Figure 1B). This indicates that mechanisms affecting pre-mRNA levels do not lead to a substantial change in the final ratio of spliced/un-spliced mRNAs as long as the pre-mRNA remains a potential substrate for the splicing machinery, since a change at the pre-mRNA level leads to an equivalent change at the mature mRNA level and, therefore, does not affect the ratio. The estimates of differential stability generated in this study therefore represent mostly the effect of change in degradation occurring at the mature mRNA levels.”*

2. Is not there any association between the degree of the disruption of the mRNA stability control and the clinical appearances of the cancers, such as the stage or prognosis of the patients?

We have now included an analysis of the association between degree of disruption of mRNA stability and the clinical features. First, by including all tumours in a single analysis that takes into account cancer-specific differences in survival, we found that tumours with higher number of disrupted transcripts are often associated with lower survival. Secondly, as shown in the new Figure 2C, the higher the number transcripts with abnormal stability, the worse the prognosis in most cancer types. However, this is not always the case, with LUSC and KIRC showing the opposite trend, highlighting the added value of studying the association between the stability of individual mRNAs and clinical features (as in Figure 3).

These results are now included in the manuscript:

Pages 6-7: *“Interestingly, across TCGA samples, the degree of stability dysregulation, calculated as the number of differentially stabilized mRNAs per patient, was associated with reduced disease-free survival (log hazard ratio of 0.36, $P < 0.005$, using Cox proportional-hazards model correcting for the confounding effect of patient age, sex, tumour purity and cancer type). Per-cancer-type associations were also mostly positive (Figure 2C), indicating that a greater disruption of mRNA stability is overall associated with worse patient outcomes.”*

3. I wonder in how many cases a genomic mutation resided in a RNA binding protein or an miRNA or their target sites were found to be responsible for the observed disruption as the easiest case?

We agree with the reviewer that genomic mutations may drive disruptions in stability programs. However, as these events are not frequent and there are often not enough samples containing a mutation for an RBP of interest in a cancer dataset for example, identification of the specific effect of mutations of RBP/miRNA is often challenging. Furthermore, we observed a strong correlation between the expression of RBPs and their regulon activity in cancer (Figure 4D), suggesting that disruption in RBP expression (and potentially miRNA expression) is most likely the dominant mechanism underlying their changes in activity, rather than mutations. We now explicitly discuss these points in the manuscript:

Page 17: *“We note that different mechanisms may underlie the observed disruption in the RBP/miRNA regulons in cancer, including changes in the expression levels of these regulatory factors, mutations, post-translational modifications in the case of RBPs, disruption of miRNA biogenesis, competition/cooperation with other regulatory factors, and enhanced/restricted access to binding sites on target transcripts. However, at least in the case of RBPs, we observed a strong correlation between their expression and regulon activity in cancer (Figure 4D), suggesting that disruption of the expression of RBPs is most likely the dominant mechanism underlying the dysregulation of their regulons.”*

Minor points:

4. It would be helpful if the IGV view of a typical case is presented, which represent how the tag distribution changes and where the binding site of RNA binding protein is located.

We thank the reviewer for this suggestion; we have now included a new Supplementary Figure S5, showing the tag distribution for FUT9, a high-confidence RBFOX1 target that is destabilized in cancer. This example shows minimal difference in intronic coverage between tumor and normal samples, while the exonic signal is greatly reduced in tumors.

5. I wonder if there are any interpretable cases for which DiffRAC failed to give a precise inference? For example, occurrence of cancerous alternative splicing, which may trigger a cancer immune response against the aberrant protein may make an influence?

We agree with the reviewer that aberrant alternative splicing may affect the exonic read profiles in tumors, and have taken measures to limit the confounding effect of such cases. These measures, which follow the procedure previously described by Alkallas et al. (2017), are now added to the Methods section (pages 23-24). Specifically, we have focused exclusively on constitutive exons, i.e. exons that are retained in all isoforms of each gene (see the TCGA RNA-seq data processing section of the Methods) in order to ensure differential isoform length does not affect stability measurements. We have also limited our regulon analysis to transcripts that do not have alternative UTRs.

6. Is not there any association of the changes in the RNA stability and the pattern of the cancer driver mutations? Disruption of some pathways may be prone to invoke errors of post-transcriptional controls.

We believe studying the effect of classical cancer driver mutations on post-transcriptional regulation may not be directly related to the main topic of this manuscript. Also, as some of these mutations may be associated with specific cancer subtypes, it becomes difficult to establish a direct causal relationship between such mutations and post-transcriptional differences. Nonetheless, we agree with the reviewer that studying the association between somatic genetic changes and post-transcriptional pathways is interesting. During this revision, we attempted to look at the association between the 'degree of disruption' of RNA stability, as described in response to point #2, and the mutation status of genes known to be involved in the regulation of mRNA decay. We first attempted to look at the association between the number of destabilized genes in each tumor (compared to normal samples, as described in point #2) and the mutation status of these RNA degradation genes (i.e., whether an impactful mutation with CADD score >30 could be found in the gene in a given sample). We restricted our analysis to genes that had impactful mutations in at least 20 tumor samples in at least one cancer type. However, there were only two genes passing this threshold. As an alternative approach, we considered investigating the association between the degree of disruption of mRNA stability and allelic loss of the RNA decay genes (instead of mutation status). The rationale was that copy number loss events are more frequent than deleterious point mutations, increasing the statistical power. However, as shown in the new Supplementary Figure 14, we were not able to identify any significant associations.

7. As far as I understand, the RNA data used for the input is a mixture of various cell types in cancers. To ensure observed phenomena is the change occurring in cancer or pre-cancerous epithelial cells, it is important to also refer to the information of the surrounding normal cells. Such a "deconvolution" analysis may be possible also utilizing single cell catalog datasets. Also, validation analyses for the deconvolution results on some representative cell types would be also useful.

As the reviewer has suggested, we have performed additional analyses to deconvolve the stability changes stemming from (pre-)cancerous cells. We specifically focused on stage-associated stability changes, given that variable cell composition is a major confounder in

analysis of differences across tumours. We used a model similar to what has been recently used by the GTEx studies (Kim-Hellmuth et al., Science 2020, PMID 32913075) to separate the changes occurring specifically in the cancerous (or pre-cancerous) cells from those occurring in the other surrounding cells (schematically shown in the new Figure 3D). The results of this deconvolution analysis are presented in the new panels E-H of Figure 3. As shown in this figure, most of the stage-associated stability changes that we had previously identified appear to stem from malignant cells, further supporting the notion that cancer cells undergo stability remodeling during disease progression. These findings are now described in the Results section “*DiffRAC identifies cancer-associated changes in mRNA stability*” (page 9), with the associated methods described in the new Methods section “*Deconvolution of cellular origin from differential stability estimates*” (pages 24-25).

Reviewer #2 (Remarks to the Author):

Key results

The authors developed and employ a statistical framework, DiffRAC, to assess the contribution of RNA stability to the transcriptome composition of 18 cancer types from RNAseq data deposited in TCGA. They then benchmark these computationally inferred changes in stability by using a previously generated benchmarking dataset for RNA stability. The authors then examine the RNA stability differences in a pair of highly and poorly metastatic breast cancer cell lines using DiffRAC and compare these results to previously generated experimental stability data from 4-thiouridine pulse-chase experiments in these same cell lines. The authors observe a concordance between DiffRAC-derived and experimentally derived changes in transcript stability, and also observe these changes in normal vs. breast tumor samples. More broadly, the authors also observe cancer-type specific associations between transcript stability and tumor stage and/or grade. Next, the authors examine the role of RNA binding proteins in regulating these stability changes. They find associations between different RBPs and their binding motif-based regulons that are cancer-type specific. They then more closely examine the RBFOX proteins and their role in regulating RNA stability in GBM. Using several previously generated datasets, the authors conclude that RBFOX1 and 3 regulate RNA stability in GBM and specifically regulate calcium signaling via stabilization of transcripts. Finally, the authors examine the role miRNAs may play in modulating the observed changes in RNA stability. They find widespread changes in the stability of miRNA regulons across cancer types. They then examine the role of miR-29 in regulating RNA stability in KIRC, and perform a miR-29 gain-of-function experiment in one KIRC cell line, finding that this does result in destabilization of the miR-29 regulon.

General comments

This paper provides a valuable resource for examining RNA stability changes in human cancer samples for which experimental stability measurements cannot be obtained. However, to support the hypotheses based on their computationally inferred stability data, the authors rely heavily on analysis of experimental data generated by other groups, some of which has limited relevance to their hypotheses, and does not provide entirely novel results (ie, the role of miR-

29 in cancer is well-documented, see review Kwon et al 2018). It would strengthen their study to provide additional experimental data to support these claims, especially with regard to the role of RBFOX proteins regulating transcript stability. The DiffRAC analysis itself is novel and will be of interest to the community, but the authors must be careful to not overstate the conclusions generated by their fairly cursory analysis of the downstream regulators of RNA stability across cancers.

We thank the reviewer for their positive assessment regarding the novelty of our analysis. As outlined below, we have now performed additional analyses and experiments to further support our claims and address the concerns of the reviewer.

Major points

1. It is not entirely clear which benchmarking data the authors used for testing DiffRAC. The paper the authors reference contains many datasets, but the authors only show the results from one: mouse ESCs vs differentiated neurons. Please clarify which data was used for benchmarking, and if it was the single dataset please reference the original paper where this data was generated. It is also unclear as to what the authors mean by “ground-truth transcript half-life measurements”, is this referring to experimentally generated half-life measurements? On a technical note, if these measurements were made using actinomycin D treatment, as I infer from the text, this is a less accurate method for measuring transcript stability than many of the newer metabolic labeling methods. It may be more useful to benchmark their framework against data generated using a more accurate method of measuring transcript stability. In either case, the method used for measuring transcript stability in their benchmarking data needs to be clearly stated in the text and the figure legend.

We thank the reviewer for pointing out that the original paper describing the stability measurements was not cited. We have now included this citation in the manuscript. We have also clarified in the manuscript that by ground-truth measurements, we mean experimental half-life estimates based on time point measurements of mRNA abundance:

Page 5: “We evaluated the performance of DiffRAC for estimating differential mRNA stability using a previously published dataset (Tippmann et al., 2012), consisting of RNA-seq data from mouse embryonic stem cells and terminal neurons, along with experimentally measured transcript half-life measurements after transcriptional blockage with actinomycin D, which here we consider as ‘ground-truth’ measurements for benchmarking purposes.”

We agree with the reviewer that transcriptional blockage with actinomycin D introduces unwanted effects that may confound the mRNA stability measurements. To address this issue, we have now added another benchmarking analysis using data from a study by Arango et al. (2018) (GEO accession GSE102113), where the less invasive BRIC-seq measurements were obtained from NAT10 knockout HeLa cells. Similar to our previous observations on the dataset

from Tippmann et al., we saw that for genes with narrow confidence intervals for both the DiffRAC estimates and the BRIC-seq-based stability measurements, the two methods produce correlated estimates. These results are now shown in Supplementary Figure S3. These results are mentioned in the section *“A generalized linear model for statistical testing of mRNA stability”*. We also describe the dataset and analysis in a new Methods section *“Differential RNA stability between NAT10 knockout and parental cells”*.

2. Please specify the RNAseq method(s) that were used for sequencing the samples in TCGA. How many datasets came from poly(A) selected RNA vs rRNA depleted RNA? If any of the sequencing data was from poly(A) selected RNA are all of the normal vs tumor samples subjected to the same RNA selection method? I assume that the two methods would result in different proportions of intronic vs exonic reads, therefore affecting DiffRAC output.

We have now clarified this point in the manuscript, Methods section *“TCGA RNA-seq data processing”*:

Page 23: *“All TCGA RNA-seq data used in this study was generated from poly(A)-selected RNA”*.

We have also added a brief discussion of the effect of different RNA-seq methods:

Page 20: *“Different RNA selection methods can also affect the intronic read counts. Poly(A)-selected RNA will lead to a lower proportion of intronic reads compared to rRNA-depleted RNA. In the current study, we made use of several poly(A)-selected datasets, including the RNA-seq data from TCGA. However, since all samples in each dataset were analysed using the same method, the estimates are all affected in a similar manner across the sample types and cancer types. We note that poly(A)-selected RNA has previously been shown to produce sufficient intronic reads for stability estimation (Gaidatzis et al., 2015). In addition, the large number of samples included in this study most likely mitigates any statistical power loss that results from lower amount of intronic reads”*.

3. The authors find that MDA-LM2 vs MDA-MB-231 stability changes are correlated with those in BRCA vs normal tissue in the TCGA data, with some overlap in the transcript sets. However, since the MDA-LM2 and MDA-MB-231 comparison is a model for metastasis (the MDA-MB-231 line is already considered a metastatic cell line) it would be more informative to look at the correlation between measured differential transcript stability in the MDA-LM2 vs MDA-MB-231 cells with stability changes across breast tumor stage and/or grade. Moreover, it would be interesting to look at this by breast cancer subtype as the MDA-MB-231 line is a triple negative line, and stability changes may be specific to this subtype.

As suggested by the reviewer, we performed an analysis to infer stability changes associated with stage in the TCGA-BRCA dataset only for TNBC patients, correcting for the confounding effects of age and purity. However, we did not have enough statistical power to detect

significant changes (we detected only one), most likely due to the small number of TNBC patients in this dataset and/or the heterogeneity among the samples. We therefore studied an alternative model, namely patient-derived xenograft (PDX) models, which have been previously used to study mechanisms of metastasis in breast cancer (Fish et al., Science 2021). These samples may in fact be a more relevant model for metastasis compared to studying cancer stage. By applying DiffRAC to RNA-seq data from two highly metastatic PDXs and a poorly metastatic PDX, we estimated the differential stability between these conditions and compared to timeseries-based differential stability estimates between MDA-LM2 and MDA-MB-231. As shown in the new Figure 2F, we observed a good agreement between the two datasets, with mRNAs that are more stable in the invasive MDA-LM2 cell line being also overall more stable in the highly metastatic PDXs compared to the poorly metastatic PDX, and similarly for destabilized mRNAs.

We have included these new results in the manuscript Results section *“DiffRAC identifies cancer-associated changes in mRNA stability”*, along with the new Figure 2F.

4. The authors find stage and grade-associated changes in RNA stability according to tumor type. Please explain how this analysis was done- for cancer types with multiple stages or grades do the RNA stability change associations reflect a consistent increase or decrease in stability across tumor stage or grade (ie, least stable in stage 1, intermediate stability in stage 2 and highest stability in stage 3)? Or is it rather stability changes between any of the stages or grades?

We have now added additional information to the manuscript to clarify our stage/grade analysis. Specifically, on page 8: *“To understand whether normal-to-tumour stability changes are correlated with progression-associated stability changes across other cancers, we used DiffRAC to examine the effect of tumour stage and grade on mRNA stability in each TCGA cancer type, by including stage/grade (as numerical variables) in DiffRAC’s GLM design while controlling for the confounding effects of age, sex and tumour purity (Supplementary Table S4). The differential stability results therefore reflect the change in stability that occurs as tumour stage or grade increases”*.

Since stage/grade are encoded as numerical variables, the identified mRNAs represent those with more or less consistent increase/decrease in stability as tumor stage or grade increases.

5. The authors’ case for RBFOX proteins playing a role in RNA stability in GBM, while intriguing, is limited by their reliance on previously generated data. Some of the data they analyze to support their hypothesis comes from experiments on systems different than the one they’re making their claim about. Specifically, to generate their “high-confidence stability network” the authors use Rbfox CLIP data collected from whole mouse brain tissue. This is good quality CLIP data, but human and mouse transcriptomes are not identical, and this limitation should be

addressed in the text. To make a more convincing claim that this CLIP data mimics RBFOX binding in human neuronal tissue the authors could examine the conservation between the 3'UTRs bound by Rbfox in mouse with their human homologs.

As suggested by the reviewer, we now show in Figure 5C that the sequence of the Rbfox binding sites identified based on mouse CLIP data are conserved in human (with the RBFOX motif preserved in most cases). Also, as explained below in response to comment 7, we have performed phenotypic activation of RBFOX1 in a human cell line model of GBM, and show that the high-confidence targets identified based on mouse CLIP data are responsive to RBFOX1 overexpression in human cell lines, further highlighting their conserved function in human.

6. I could not find details how the authors' generated their Rbfox high-confidence stability network besides that it was based on a specific CLIP data set and involved 3'UTR binding with height greater than 200. Please specify a unit for 200, and also consider using the CIMS-called peaks from this CLIP data set as this indicates more robust binding than simple number of reads. Also specify data from which Rbfox CLIP experiments was used. The paper referenced performed CLIP on Rbfox1,2 and 3. Please elaborate on this in the methods or in the text.

We thank the reviewer for pointing out that we had not included this information in the Methods. This information is now added to the manuscript (Methods section "*HITS-CLIP data analysis*"):

Pages 26-27: "Pooled HITS-CLIP peaks of RBFOX1/2/3 proteins in whole brain tissue lysate of mice were retrieved from a previous study (Weyn-Vanhentenryck et al., 2014). Peaks occurring in the 3'UTR with a height greater or equal to 200 overlapping CLIP tags were retained (peak height was extracted from Supplementary Table 1 of the source publication). The mRNAs that had at least one 3' UTR high-confidence peak were considered high-confidence RBFOX targets, which were further filtered to include only those whose orthologs had expression measurements in TCGA. This resulted in 58 genes, 54 of which also have a 3' UTR RBFOX binding site based on CIMS analysis of CLIP data".

As noted above, CIMS-called peaks also support 93% of the mRNA targets identified based on peak height at our selected threshold.

7. The authors present their analyses of loss-of-function data from differentiated primary human neural progenitor cells with RBFOX1 knockdown as well as gain-of-function data from mouse neurons with RBFOX1 knockout in which RBFOX1 expression has been restored. In both cases the authors are examining the function of RBFOX1 in non-transformed neuronal models (and in one case, again, a mouse model); however, their hypothesis points to a role for RBFOX1 in human tumorigenesis. Therefore, stronger experimental evidence is needed to make a claim about the role of RBFOX proteins role in regulation of transcript stability in human cancers.

Ideally this would include overexpression of RBFOX1/3 proteins in human glioblastoma cells lines followed by RNA-seq for DiffRAC analysis, as well as experimental transcript stability measurements in these cell lines. Similar loss-of-function experiments in human glioblastoma cells lines would likewise make their conclusions much stronger. With the data presented I would also caution the authors to be careful about over-interpreting any GO terms enriched in the overlapping genes from the datasets they've analyzed, as they combine data from a variety of non-cancer and non-human experimental systems. Unless they provide further experimentally based evidence (from human glioblastoma cell lines) the authors do not convincingly make their claim about the role of the RBFOX regulon in calcium signaling in human cancers.

To address the issue raised by the reviewer, we have now added new experimental data from overexpression of RBFOX1 in the human glioblastoma cell line A172 followed by RNA-seq (in comparison to GFP control). As shown in the new Figures 5H-K, phenotypic activation of RBFOX1 by ectopic expression results in widespread gene expression changes, with CLIP-based RBFOX1 targets specifically being up-regulated as expected. This experiment also replicated our observation that calcium signaling genes are positively regulated by RBFOX1, and showed that by rescuing RBFOX1 expression we can up-regulate a substantial portion of the pan-cancer-destabilized RBFOX targets that we had identified in the previous version of the manuscript. These results support our previous conclusions about the role of RBFOX1 in regulation of mRNA stability and abundance in GBM (and potentially other cancers).

These observations are discussed in the Results section "*RNA-binding proteins play a key role in shaping the tumour mRNA stability profile*" (page 12), in addition the new Figures 5H-K.

We note that we opted to perform over-expression experiments as opposed to loss-of-function experiments given that RBFOX1 is already expressed at extremely low levels in the majority of GBM cell lines.

8. The authors provide basic experimental evidence that miR-29 is sufficient to downregulate transcripts that harbor miR-29 target sites in one KIRC cell line, and that there is significant overlap between these transcripts and those downregulated in the TCGA KIRC tumor vs normal data. Without further experimental evidence though the authors should not claim that miR-29 downregulation has a widespread effect on the stability of genes in cancer. If the authors would like to support this claim, they should perform miR-29 overexpression and inhibition experiments across multiple human KIRC cell lines, or across multiple lines from cancers in which the authors found particularly robust evidence for miR-29 regulation.

We have performed additional miR-29 overexpression and inhibition experiments in two different KIRC cell lines. More specifically, we have expressed a miR-29 mimic in the A-498 cell line model of KIRC, and used a miR-29 inhibitor in the ACHN cell line. As shown in the new Figure 6E and Supplementary Figures S9-10, the results are in agreement with our previous

conclusions: miR-29 overexpression down-regulated miR-29 targets (which are up-regulated in KIRC), and miR-29 inhibition has the opposite effect. Importantly, of the 69 genes previously presented in Figure 6 as being up-regulated in KIRC potentially due to miR-29 inactivation, 53 (~77%) showed significant differential expression in the expected direction after miR-29 overexpression in the new A-498 cell line and inhibition in ACHN cell line, suggesting the reproducibility of our observations across different experimental models. These genes are now highlighted in the new Figure 6F-G.

Minor points

1. Many times throughout the text the authors refer to stabilized genes. To avoid confusion please refer to stabilized transcripts or mRNA when referencing RNA.

We thank the reviewer for pointing this out. The correction was made.

2. Related to figure S3, is the variation across cancer types in the number of transcripts with stability changes passing the statistical threshold related to the number of samples analyzed per cancer type or is there another explanation for this variability? Also, I believe a number has been misreported in the COAD graph in this figure, which should be fixed.

We thank the reviewer for pointing out the incorrect number in the COAD plot, which should be 2812. We have made the appropriate modification (the figure number is now S4).

Regarding the variation in the number of differentially stabilized genes among cancer types, indeed, we see a correlation between the number of samples and the number of differentially stabilized genes, suggesting varying degree of statistical power. This point is now shown in the new Supplementary Figure S12 and also discussed in the manuscript:

Page 14: *“We note that about 57% of the variability in the number of differentially stabilized genes across cancer types appears to be attributed to sample size, suggesting that our analysis may be underpowered for smaller cancer cohorts (Supplementary Figure S12)”*.

3. Is figure 2f simply a graph of data from Goodarzi et al? This figure seems superfluous.

This figure (which is now panel i of Figure 2) plots the reprocessed data from Goodarzi et al (and not the data directly reported in that paper). This reprocessing involves read mapping, counting, variance-stabilized transformation, and estimation/removal of cell-specific differences in baseline expression, as discussed in the Methods section and now in the legend of Figure 2. Therefore, we believe its inclusion provides added value, especially to visually convey the message presented by means of statistics in previous panels. As discussed in the

Methods section “*Differential RNA stability between MDA-MB-231 and MDA-LM2 cells*”, we have used a model that is aware of the distributional properties of RNA-seq reads in order to fit a model for time-dependent transcript decay to the data from Goodarzi et al. — Figure 2i shows example genes that are significant based on this model; we believe such visualizations are useful to examine whether the statistical model in fact captures the behaviour that we are interested in.

4. In figure 3b please choose colors with better contrast to better visualize the datapoints.

We have increased the contrast in Figure 3B.

5. Please provide an explanation for identifying RBFOX1 and RBFOX3 as regulators of the observed effects on RNA stability but excluding RBFOX2. All the RBFOX protein are highly similar, and also share the same binding motif. Furthermore, there is a fairly robust RBFOX2 CLIP dataset from the Yeo lab’s eCLIP study, which may be complementary to the CLIP data the author’s have used, as it is from human cell lines.

We have now added an explanation of the reason we have focused on RBFOX1 and RBFOX3:

Page 15: “*We note that the RBFOX family of proteins includes RBFOX1, RBFOX2, and RBFOX3; however, RBFOX1 and RBFOX3 show the greatest extent of down-regulation across different tumours (>60-fold, Figure 5A-B), whereas RBFOX2 shows comparatively moderate down-regulation (~3-fold, Supplementary Figure S13). Furthermore, RBFOX2 does not show significant correlation with the expression of the mRNAs that contain the RBFOX-binding consensus sequence (Ray et al., 2013). Taken together, these observations suggest that RBFOX1/3 are the most likely candidates driving dysregulation of the RBFOX regulon in cancer*”.

As suggested by the reviewer, we also considered eCLIP data for RBFOX2, but we have concluded that this dataset is not suitable for our analysis here. First, as described above, RBFOX2 is not significantly down-regulated in GBM, and therefore its eCLIP data may not be directly related to the biological system that we are studying. Secondly, we believe CLIP data from (mouse) brain tissue may be more relevant to studying GBM than the eCLIP data that Yeo’s lab (obtained from HepG2 or K562 cell lines, which represent drastically different tissues). Furthermore, when examining the RBFOX2 eCLIP data, we could not see any evidence of RBFOX2 binding to the 3’UTR of many of our high-confidence RBFOX target genes (such as SHANK1, CAMK4, PRKCB). This is in contrast with mouse studies showing that RBFOX2 binds to similar sites as RBFOX1/3, which most likely suggests that the RBFOX1/3-regulated transcripts, which are the focus of Figure 5, are not expressed in HepG2 or K562 cells. Furthermore, as we discussed above, we have found that RBFOX targets identified based on the mouse CLIP data are highly responsive to RBFOX1 ectopic expression in GBM cell line models, leading us to conclude that the targets we have identified based on mouse CLIP data are in fact relevant to the study of GBM.

7. In figure 4d it may be helpful to highlight the datapoints representing any of the RBFOX related points, as this is a visually busy graph and RBFOX is the regulon the authors focus on.

We have now highlighted the RBFOX1 regulon in Figure 4D.

8. The miR-29 mimetic and control RNA sequences used for the transfection experiment should be reported.

We have now added this information to the Methods section:

Page 27: "For transient transfection, 786-O and A-498 cells (100,000 cells/well in 6-well plates) were reverse-transfected in antibiotic-free medium with 10 nM of miRNA-29 mimic (stem-loop sequence:

UGGUUUCGUAUUGGUGCAUAGAAGUAUUAAUUUUGUAACUUGUCUAGCACCAUUUGAAACCAGU , mature miRNA sequence: UAGCACCAUUUGAAACCAGU, ThermoFisher, 4464066) [...]"

9. In figure S5 it would be helpful to also include a graph showing the relative quantitation values of miR-29 expression relative to the control, along with the graph of the raw Ct data. This will make it much easier to look at relative fold-change in expression of the miRNA.

We have now added a new panel to this figure as suggested by the reviewer (please note that the new number for this figure is S8).

REVIEWERS' COMMENTS:

Reviewer #1 (Remarks to the Author):

First of all, I appreciate the efforts of the authors to revise the manuscript. Thanks to the extensive analyses and the deepened discussion, I think this manuscript has been very much improved. The comments I have made in the previous round of the review have been mostly adequately addressed. I sincerely hope that this paper should pave the first step towards making better use of the mRNA quality control mechanism for improved diagnosis and treatments of the patients.